# *MOESART*: An effective Sampling-based Router for Sparse Mixture of Experts

## Abstract

The sparse Mixture-of-Experts (Sparse-MoE) is a promising framework for efficiently scaling up model capacity. This framework consists of a set of experts (subnetworks) and one or more routers. The routers activate only a small subset of the experts on a per-example basis, which can save on resources. Among the most widely used sparse routers are Top-$k$ and its variants, which activate $k$ experts for each example during training. While very effective at model scaling, these routers are prone to performance issues because of discontinuous nature of the routing problem. Differentiable routers have been shown to mitigate the performance issues of Top-$k$, but these are not $k$-sparse during training, which limits their utility. To address this challenge, we propose *MOESART*: a novel $k$-sparse routing approach, which maintains $k$-sparsity during both training and inference. Unlike existing routers, *MOESART* aims at learning a good $k$-sparse approximation of the classical, softmax router. We achieve this through carefully designed sampling and expert weighting strategies. We compare *MOESART* with state-of-the-art MoE routers, through large-scale experiments on 14 datasets from various domains, including recommender systems, vision, and natural language processing. *MOESART* achieves up to $16\%$ (relative) reduction in out-of-sample loss on standard image datasets, and up to $15\%$ (relative) improvement in AUC on standard recommender systems, over popular $k$-sparse routers, e.g., Top-$k$, V-MoE, Expert Choice Router and X-MoE. Moreover, for distilling natural language processing models, *MOESART* can improve predictive performance by $0.5\%$ (absolute) on average over the Top-$k$ router across 7 GLUE and 2 SQuAD benchmarks.

## 1 Introduction

Scaling up model capacity is a promising way to achieve better performance on a wide range of problems such as language model pre-training (Radford et al., 2019; Raffel et al., 2020; Lewis et al., 2020; Lample & Conneau, 2019), and visual representation learning (Dosovitskiy et al., 2021; Bao et al., 2022). However, high capacity in neural networks typically requires significant computational resources during both training and inference, which may be prohibitive. This computational requirement is mainly due to the dense nature of computation in neural networks, i.e., a model uses all its parameters to process each input example. To address this limitation, sparsely activated models present a viable solution (Bengio et al., 2016; Shazeer et al., 2017; Chen et al., 2020; Zuo et al., 2022a). Sparsely activated models use a subset of parameters to process each input.

The Sparse Mixture of Experts (Sparse MoE) (Shazeer et al., 2017) is a promising framework for obtaining sparsely activated models. Sparse-MoE consists of a set of trainable experts (neural networks) and a trainable router. The router adaptively selects an appropriate subset of experts on a per-input basis during model training and inference. This adaptive selection makes it possible to train Sparse MoE models that are orders of magnitude larger than densely activated models without significant increase in training costs (Shazeer et al., 2017; Lepikhin et al., 2021; Fedus et al., 2022). In addition, these sparsely activated models can learn faster than their compute-matched dense models. For example, Switch Transformers (Fedus et al., 2022), based on the Top-$k$ router, can learn $7\times$ faster than dense T5 model (Raffel et al., 2020)—the number of FLOPs used per-input during training and inference is the same for both models. Such Sparse MoE models have shown state-of-the-art performance in natural language processing (NLP) (Shazeer et al., 2017; Fedus et al., 2022; Du et al., 2022; Artetxe et al., 2022), vision (Ruiz et al., 2021), and distillation (Zuo et al., 2022b).

The choice of router plays a central role in Sparse-MoEs. For efficient training, the literature in Sparse-MoEs has been primarily based on Top-$k$ routing (Shazeer et al., 2017), which selects $k$ out of $n$ experts using a top-$k$ operation. Top-$k$ routing is simple and efficient because it allows conditional training: in backpropagation, for each input example, only the gradients of the loss with respect to $k$ experts need to be computed. With a customized implementation, conditional training can lead to large computational savings. There have been various follow-up works that have proposed variants of Top-$k$ routing (Ruiz et al., 2021; Lepikhin et al., 2021; Zhou et al., 2022; Zoph et al., 2022; Chi et al., 2022). These approaches fundamentally rely on a top-$k$ operation to exploit conditional training. Prior literature (Ruiz et al., 2021; Hazimeh et al., 2021; Fedus et al., 2022) has highlighted performance and stability issues with Top-$k$ routing, primarily attributing it to the discontinuous nature of Top-$k$ operation, which implies that gradient does not exist at certain inputs. Hence, some differentiable routing approaches (Hazimeh et al., 2021; Ibrahim et al., 2023; Sander et al., 2023) were proposed to mitigate these issues. However, these approaches are more expensive per training step as they require gradients with respect to more than $k$ experts.

In this paper, we focus on improving $k$-sparse routing in Sparse MoEs. Specifically, we propose *MOESART* (**M**ixture **o**f **E**xperts with **SA**mpling based **R**ou**T**ing, inspired from *Mozart* and pronounced "mow-saart"): a novel sampling-based approach for conditional training in Sparse MoEs. We model the routing function as a learnable parameterized softmax distribution. To achieve sparsity, we sample $k$ times from the parameterized distribution, identifying the selected experts per-input. We then assign weights to the selected experts; these weights are adjusted to ensure that the resulting prediction is a good approximation of the standard softmax router. Although the sampling process is itself non-differentiable, the adjustment strategy on the weights ensures a gradient is passed to the parameterized router distribution, in particular to the elements associated with the selected expert indices. In expectation, the adjusted router weights serve as a good sparse approximation of the router probabilities of the dense softmax.

Although Top-$k$ is considered to be a $k$-sparse approximation of the classical (dense) MoE models (Clark et al., 2022), we empirically show that our $k$-sparse approximation approach appears to be superior than Top-$k$ based approximations for learning routing. The greedy nature of Top-$k$ leads to a biased estimation strategy. Top-$k$ routing can potentially suffer from selection bias (Ovaisi et al., 2020), where a suitable expert is ranked too low by the router for an input example to be sufficiently exposed to during training. This approach may ignore potentially informative experts whose contribution is overshadowed by the top-$k$ operation. Consequently, it might not fully leverage the diversity and richness of the expert pool, while optimizing the combinatorially challenging routing problem. In contrast, our approach can mitigate this selection bias as any expert can be selected because of sampling during the course of training. Our experiments indicate that *MOESART* outperforms the Top-$k$ router and many of its variants on datasets from various domains: vision, recommender systems, and NLP.

**Contributions.** Our technical contributions can be summarized as follows:

- We propose *MOESART*: a novel $k$-sparse routing approach, which maintains $k$-sparsity during both training and inference, hence allowing for conditional training and inference. Conditional training allows sparse backpropagation, where for each input example, only the gradients of the loss w.r.t. $k$ experts need to be computed. This is similar to Top-$k$ routing (Shazeer et al., 2017).
- Unlike existing routers, *MOESART* aims at learning a good $k$-sparse approximation of the classical, softmax router (Jacobs et al., 1991). We achieve this through sampling and expert reweighting strategies. We empirically show that our sampling-based training approach learns a better sparse approximation of the classical (dense) MoE models than Top-$k$ style routers.
- On standard recommender systems and vision datasets, *MOESART* substantially improves performance of MoE models over several SOTA $k$-sparse routers: Top-$k$ (Shazeer et al., 2017), V-MoE (Ruiz et al., 2021), Expert Choice Router (Zhou et al., 2022), and X-MoE (Chi et al., 2022) .
- In distillation of pre-trained natural language processing models, *MOESART* consistently improves over Top-$k$ in distilling BERT into MoEBERT on 7 GLUE and 2 SQuAD benchmarks.

## 2 RELATED WORK

The MoE framework was introduced by Jacobs et al. (1991). More recently, Shazeer et al. (2017) proposed a *Sparse*-MoE framework which routes each input to a subset of experts and showed good

performance on NLP tasks. This sparked off a series of works on routing strategies in the Sparse-MoE paradigm. These can be categorized and summarized as follows:

- *Sparse routing.* Sparse routers exactly activate a user-specified number of experts in each training step on a minibatch of inputs. Shazeer et al. (2017) proposed Top-$k$ routing, which selects $k$ experts per input. Ruiz et al. (2021) reordered softmax and top-$k$ operation in Top-$k$ router and showed better performance on vision tasks. Chi et al. (2022) proposed to compute routing scores on a low-dimensional hypersphere. Zhou et al. (2022) proposed to let experts select top inputs. Recently, Sander et al. (2023) proposed a differentiable relaxation of the Top-$k$ operator. Their smoothing approach does not always guarantee exact $k$-sparsity. It also requires solving an optimization subproblem per-input during inference, which can be more costly.
- *Routing as assignment.* Lewis et al. (2021) and Clark et al. (2022) formulate routing as an assignment problem using linear programming and optimal transport for balanced routing. Liu et al. (2023) proposed an optimal transport formulation with support for $k$-sparsity constraints. All these approaches are also sparse but more expensive than Top-$k$ style routers.
- *Dense-to-Sparse routing.* Hazimeh et al. (2021) and Ibrahim et al. (2023) propose differentiable routers, which improve over Top-$k$ in terms of stability and statistical performance. Although these routers can allow conditional inference, they are less appealing in terms of conditional training. These routers can only partially allow for conditional training (during a later stage of training with customized implementations) and are thus more expensive during training.
- *Randomized routing.* Some works propose randomized routing strategies to bypass learning the routing function. Roller et al. (2021) use hashing to randomly assign inputs to different experts *before* training. Zuo et al. (2022a) routes all samples in a minibatch to two randomly chosen experts and forces experts to produce similar predictions. These methods cannot be technically categorized as MoE models as these do not learn a routing parameterization.

The goal of our work is to propose a new $k$-sparse routing approach such that the per-minibatch compute cost is the same as Top-$k$ style routers. In contrast to the above routing approaches, our approach is the first to explore the idea of sampling in Sparse-MoE.

A related line of work focuses on stochastic $k$-subset selection in non-MoE settings. Paulus et al. (2020); Chen et al. (2018); Xie & Ermon (2019) propose differentiable methods for sampling $k$-subsets from a categorical distribution, based on generalizations of the Gumbel-softmax trick (Maddison et al., 2017; Jang et al., 2017). If these methods were to be applied to Sparse-MoE, they would perform dense training (i.e., the gradients of all experts, even if not selected, will be computed during backpropagation), hence limiting their usefulness for conditional computing. In contrast, *MOESART* exploits sparsity for conditional training.

## 3 ROUTING IN MIXTURE OF EXPERTS

We first review the classical MoE learning paradigm. We assume that the task has an input space $\mathcal{X} \subseteq \mathbb{R}^p$ and an output space $\mathcal{Y} \subseteq \mathbb{R}^u$. In the MoE framework, the prediction function has two components: (i) a set of $n$ experts (neural networks) $f_i : \mathcal{X} \to \mathbb{R}^u$ for any $i \in [n] := \{1, 2, \ldots, n\}$, and (ii) a router $g : \mathcal{X} \to \Delta_n$ that outputs weights in the probability simplex $\Delta_n = \{g \in \mathbb{R}^n : \sum_i g_i = 1, g \geq 0\}$. Given a sample $x \in \mathcal{X}$, MoE combines the expert outputs as follows: $\sum_{i=1}^n f_i(x)g(x)_i$. Recall that classical (dense) MoE minimizes the following objective:

$$\min_{\{f_i\}, g} \hat{\mathbb{E}} \left[ \ell \left( y, \sum_{i \in [n]} f_i(x)g(x)_i \right) \right],  \tag{1}$$

where $\hat{\mathbb{E}}$ denotes expectation over the training dataset $\mathcal{D} = \{(x_1, y_1), \cdots, (x_N, y_N)\}$, $\ell(\cdot)$ denotes the loss function used during training, and $g(x)$ is a softmax router, which is typically expressed as $g(x) = \text{Softmax}(Ax + b)$, where $A \in \mathbb{R}^{n \times p}$ and $b \in \mathbb{R}^n$ denote learnable router parameters.

Different from the classical MoE above, Sparse-MoEs use a router that selects a convex combination of $k$ out of the $n$ experts per-input, where typically $k \ll n$. Next, we discuss some state-of-the-art sparse routers used in sparse-MoE for parameterizing $g(\cdot)$.

- **Top-$k$**: The Top-$k$ router (Shazeer et al., 2017) is defined as $g(x) := \text{Softmax}(\text{Top}k(Ax + b, k))$, where for any vector $v$, $\text{Top}k(v, k)_i := v_i$ if $v_i$ is in the top $k$ elements of $v$, and $-\infty$ otherwise.

- **V-MoE**: Lepikhin et al. (2021); Ruiz et al. (2021) proposed to reorder the $\mathrm{Softmax}(\cdot)$ and $\mathrm{Top}k$ operation in the Top-$k$ router. Ruiz et al. (2021) parameterizes the router as follows: $g(x) := \mathrm{Top}k(\mathrm{Softmax}(Ax + b + \epsilon), k)$, where $\epsilon \sim \mathcal{N}(0, \frac{1}{n^2})$ and for any vector $v$, $\mathrm{Top}k(v, k)_i := v_i$ if $v_i$ is in the top $k$ elements of $v$, and $\mathrm{Top}k(v, k)_i := 0$ otherwise. We denote this router as V-MoE.
- **SMoE:** Fedus et al. (2022) parameterize the router in Switch Transformers with multiplicative noise as follows: $g(x) := \mathrm{Top}k(\mathrm{Softmax}(Ax\epsilon + b), k)$, where $\epsilon \sim \mathcal{U}(0.98, 1.02)$.
- **Expert Choice Router**: Zhou et al. (2022) define their sparse router on a minibatch as: $G(X_{\mathcal{B}}) := \mathrm{Top}k(\mathrm{Softmax}(AX_{\mathcal{B}} + b)^T, k')^T$, where for any vector $v$, $\mathrm{Top}k(v, k')_i := v_i$ if $v_i$ is in the top $k'$ elements of $v$, and $\mathrm{Top}k(v, k')_i := 0$ otherwise. Note that $k' = |\mathcal{B}| * k/n$, where $k$ is the (average) sparsity level per-input such that each expert $i \in [n]$ selects $k'$ samples in a batch of size $|\mathcal{B}|$.
- **X-MoE**: Chi et al. (2022) estimate the routing scores on a low-dimensional hypersphere by parameterizing the router as follows: $g(x) := \mathrm{Top}k(\mathrm{Softmax}(((APx)/(\|A\|_{L_2} \|Px\|_{L_2}))/\tau), k)$, where $P \in \mathbb{R}^{\frac{n}{2} \times p}$, $A \in \mathbb{R}^{n \times \frac{n}{2}}$, $\tau > 0$ are all learnable parameters.

Both Top-$k$ (or its variants above) and Softmax routers have their pros and cons. Top-$k$ style routers allow for conditional training, i.e., in the forward pass, for each minibatch of size $B$, only $kB$ (instead of $nB$) expert evaluations (i.e., $f_j(x)$) are required, and hence in backpropagation, only the gradients of the loss with respect to $kB$ elements need to be computed. With a careful implementation, conditional training can give significant computational savings. However, the discontinuous nature and selection bias in Top-$k$ can lead to challenges during optimization. On the other hand, the softmax router is smooth, hence can be easier to optimize. However, the softmax router can be computationally expensive during both training and inference: the router score for each expert is non-zero; hence, all experts $f_i(x)$ are used per-input $x$.

**Motivation.** Our approach tries to combine the benefits of the two approaches. We propose to learn a sparse approximation of the softmax router via sampling such that it has the following desirable properties: (i) Per-step training costs are similar to Top-$k$, which is a central consideration in Sparse MoE models. (ii) As we demonstrate in our experiments, sampling can reduce the selection bias prevalent in greedy Top-$k$ based selection approaches, especially in the early stages of optimization, reducing the likelihood of bad routing solutions.

### 3.1 THE *MOESART* APPROACH FOR CONDITIONAL COMPUTING

We formulate a sparse approximation of the classical MoE objective in Problem (1). The goal is to train a sampled version of the softmax router to approximate the softmax router during the course of training. The high-level idea can be summarized as follows. For each input $x$, we sample a subset of experts from a parameterized distribution $g(x)$, and use a modified version of $g(x)$ on the *sampled set* as a sparse approximation for conditional computation. This is shown in Fig. 1.

Let us denote the dense softmax router by the function $g(\cdot)$. For each input $x$, we sample $k$ times from the discrete set $\{1, \cdots, n\}$ with distribution $g(x)$. We define a random vector $r(x) = (r_1, \cdots, r_n) \in \mathbb{Z}_{\geq 0}^n$, which denotes the number of times each expert is sampled. We denote the unique subset of sampled experts by $s(x)$, which can be derived from $r(x)$ as follows: $s(x) := \{i : r(x)_i > 0\}$. The cardinality of set $s(x)$ is given by $|s(x)| \leq k$. For instance, as shown in Fig. 1, $s(x) = \{3, 6\}$ and $r(x) = (0, 0, 1, 0, 0, 1, 0, 0)$ represents a sample of size $k = 2$, where 3rd and 6th expert were sampled once. Similarly, $s(x) = \{1, 5\}$ and $r(x) = (1, 0, 0, 0, 2, 0, 0, 0)$, depict the case where the first expert was sampled once, and fifth expert was sampled twice. Note that $\sum_{i=1}^n r(x)_i = k$.

We seek to minimize the following objective:

$$\min_{\{f_i\}, g} \hat{\mathbb{E}} \left[ \hat{\mathbb{E}}_{r(x)} \left[ \ell \left( y, \sum_{i \in s(x)} f_i(x)\tilde{g}(x)_i \right) \right] \right], \quad (2)$$

where the outer expectation in Problem (2) is over the training set $\mathcal{D}$ and the inner expectation is computed with respect to a subset of experts in the set $s(x)$ for each input $x$. $g(x) = \mathrm{Softmax}(Ax + b)$ and $\tilde{g}(x)$ is a modified version of $g(x)$ on the set $s(x)$ — more details are given in Section 3.1.1. Additional regularization can be added to the objective, which is discussed in Section 3.2.

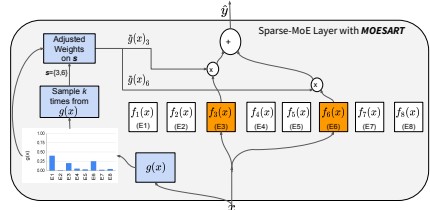

Figure 1: Sparse-MoE with *MOESART* router.

Next, we empirically show how our *MOESART* objective, without any additional regularization, learns a good $k$-sparse approximation to the dense softmax router in terms of the empirical loss, superior than that learnt by Top-$k$ router.

**MOESART is a good $k$-sparse approximation.** *MOESART* leads to a more effective sparse approximation than that learnt by Top-$k$. We empirically show this on SVHN dataset for an MoE architecture with 8 convolutional experts. The details of the architecture are outlined in Supplement Section S6.2. We trained MoE models with 3 different routing strategies: Classical MoE (softmax), Top-$k$ (sparse) and *MOESART* (sparse). We optimized with Adam (Kingma & Ba, 2015) with cross-entropy loss for 250 epochs. We set $k = 2$ for Top-$k$ and *MOESART*.

We visualize the different loss distributions on held-out data in Figure 2. In particular, we visualize the distributions, when each of the 3 routers are used to perform dense or sparse inference post-training. We also show the discrepancy between different distributions using Wasserstein Distance (Kantorovich, 1960; Peyré & Cuturi, 2020). We can see in Figure 2a that *MOESART* has a small distribution discrepancy when the model performs sparse or dense inference. In comparison, we can see a much bigger discrepancy when the model performs dense inference when the model is trained with Top-$k$ routing — see Figure 2c. We also see in Figure 2e that when the model is trained densely with classical MoE objective, a large discrepancy between loss distributions occurs when the model performs sparse versus dense inference — note that we used Top-$k$ as a sparse approximation at inference time in this setting. We also compare in Figure 2b and 2d distribution discrepancy of the loss distribution of *MOESART* and Top-$k$ with that of classical MoE. Interestingly, we see *MOESART* is closer to classical MoE loss distribution than Top-$k$. Finally, in Figure 2,

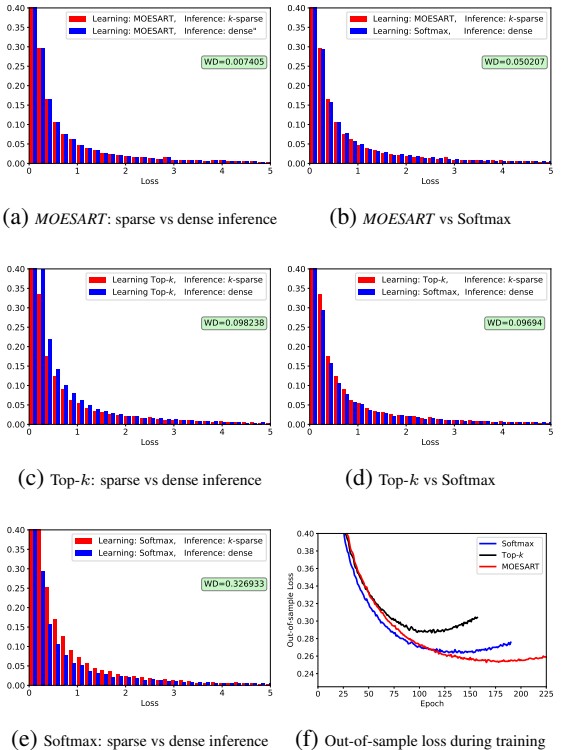

(a) *MOESART*: sparse vs dense inference

(b) *MOESART* vs Softmax

(c) Top-$k$: sparse vs dense inference

(d) Top-$k$ vs Softmax

(e) Softmax: sparse vs dense inference

(f) Out-of-sample loss during training

Figure 2: Comparison of empirical loss distributions on held-out samples from SVHN dataset when MoE models are trained with three different objectives: Softmax (classical MoE), Top-$k$ and *MOESART* (no additional regularization). The y-axis in 2a-2d is zoomed in to highlight differences between distributions. We evaluate with $k$-sparse and dense approximations at inference time for all three models. Wasserstein Distance (WD) is shown to quantify the distance between loss distributions. We can see that *MOESART* provides a good $k$-sparse approximation to its dense counterpart (see 2a) as well as to the classical MoE (see 2b). Note in 2f that classical MoE, *MOESART*, and Top-$k$ achieve held-out objective of 0.259, 0.251 (lowest) and 0.282 respectively.

we show the out-of-sample loss during the course of training. *MOESART* achieves a much smaller objective than Top-$k$; interestingly, it even surpasses the densely-trained classical MoE.

### 3.1.1 Computing $\tilde{g}$ during training and inference

Here, we outline approaches to compute $\tilde{g}$ during training and inference. $\tilde{g}$ in (2) is a modification of $g$ on the sampled indices from $g$, which serves three purposes: (i) $\tilde{g}$ is sparse as it's only non-zero on the sampled indices, hence allowing conditional computation. (ii) $\tilde{g}$ allows gradient to backpropagate to $g$ — recall that just sampling from parameterized distribution $g$ doesn't allow gradient computation with respect to router parameters in $g$. (iii) $\tilde{g}$ aims to approximate $g$ in expectation.

**Training.** We first introduce some notation. Let $\{a_i\} \in \mathbb{R}^p \ \forall i \in [n]$ and $b_i \in \mathbb{R}^n$ be learnable router parameters and $\tau > 0$ be a temperature hyperparameter. Let $o(x) = (Ax + b)/\tau$ be router logits, where $A := (a_1 \ \cdots \ a_n)^T$. Equivalently, $o_i(x) := (a_i^T x + b_i)/\tau \ \forall i \in [n]$, and $g(x)_i :=$

$\exp(o_i(x))/\sum_{j=1}^n \exp(o_j(x))$. Next, we refer to $\tilde{o}(x)$ as adjusted logits after the sampling process. $\tilde{g}(x)$ is a softmax-transformed version of adjusted logits $\tilde{o}(x)$. Note that $\tilde{g}(x)$ will be sparse with cardinality $\|\tilde{g}(x)\|_0 \leq k$. For notational convenience, we drop the dependence on $x$, and write $o(x) = o, \tilde{o}(x) = \tilde{o}, s(x) = s, g(x) = g, \tilde{g}(x) = \tilde{g}$. We first outline some natural choices for $\tilde{o}$ below and highlight their limitations.

  (i) $\tilde{o}_i := \{o_i \text{ if } i \in s\}$ or $\{-\infty \text{ if } i \notin s\}$. This corresponds to $\tilde{g}_i = g_i/\sum_{j \in s} g_j \ \forall i \in s$.
  (ii) $\tilde{o}_i := \{o_i + \log(r_i) \text{ if } i \in s\}$ or $\{-\infty \text{ if } i \notin s\}$. This gives $\tilde{g}_i = g_i r_i/\sum_{j \in s} g_j r_j \ \forall i \in s$.
  (iii) $\tilde{o}_i := \{o_i + \log(r_i) - \log(kg_i) \text{ if } i \in s\}$ or $\{-\infty \text{ if } i \notin s\}$. This gives $\tilde{g}_i = 1/k \ \forall i \in s$.

The above approaches have limitations, which make them either less appealing or ill-posed. If we first consider (i) and (ii), we get $\tilde{g}_i$ to be biased estimators of $g_i$ — see Supplement Section S3.1 for examples. If we consider (iii), the adjusted logit corrects the true logit $o_i$ by the *expected* number of occurrences of an expert $i$. This correction makes $\tilde{g}_i$ an unbiased estimator of $g_i$ — see proof in Supplement Section S1; however, the correction makes $\tilde{g}_i$ independent of $g_i$, causing gradients with respect to router parameters $A$ and $b$ to be zero — hence router parameters can not be updated.

Therefore, we propose an alternative strategy, which tends to have a smaller bias than the strategy in (i)-(ii), and has non-zero gradients with respect to router parameters unlike (iii). The idea is to follow a randomized adjustment strategy for the sampled logits $o_i \ \forall i \in s$. We randomly sample an index $z$ uniformly in the subset $s$. We propose the following adjustment strategy:

$$\tilde{o}_i := \begin{cases} o_i + \log(r_i), & i = z \\ o_i + \log(r_i) - \log((k-1)g_i), & i \in s \backslash \{z\} \\ -\infty & i \notin s \end{cases} \qquad (3)$$

This strategy tends to have a similar bias in comparison with the strategies in (i)-(ii) for uniform distribution $g$ and have a much smaller bias than (i)-(ii) for non-uniform distributions for $g$ — see ablation study in Supplement Section S3.1. Note that router probability $g$ is expected to have a range of distributions across inputs, which makes our proposed strategy less biased during training. Additionally, this approach has non-zero gradients with respect to the router parameters $A$ and $b$, allowing the router to learn unlike the unbiased strategy in (iii) above. We empirically observed that sampling without replacement performed significantly better than sampling with replacement in the context of Sparse MoE training — see an ablation study in Section 4.3. We also show that our proposed strategy in (3) significantly outperforms (i-ii) in the context of Sparse MoE training on image datasets in Table S1 in Supplement Section S3.2.

Note that all the approaches in (i)-(iii) and the one we propose require $k > 1$; for $k = 1$, these approaches have zero gradients with respect to the router parameters, this issue arises in Top-$k$ router as well as highlighted by Ruiz et al. (2021); this is why earlier works e.g., Ruiz et al. (2021); Lepikhin et al. (2021), have proposed $k > 1$, which we also follow in this work.

**Inference.** There are two important considerations in Sparse MoE models at inference time: (a) Similar to conditional training, $k$-sparse inference is crucial for efficient serving of large-scale MoE models. As highlighted by the Figure 2a, there is a small discrepancy between the loss distributions when the models perform sparse or dense inference when trained with *MOESART* objective. Hence using $k$-sparse solution works well at inference. (b) The routing should be deterministic for better interpretability and reproducibility. Therefore, we follow a $k$-sparse deterministic strategy at inference time. Instead of sampling from $g(x)$, we select the indices corresponding to the top $k$ elements of $g(x)$. We use the adjustment strategy in (iii) above — note that this leads to an equally weighted average of the top $k$ expert predictions per input. This deterministic inference strategy led to a smaller out-of-sample loss in comparison to using sampling at inference.

## 3.2 Additional Trimmed Lasso regularization

We can also add (per-input) Trimmed Lasso regularization (Bertsimas et al., 2017) in the objective (2). Trimmed Lasso regularization is defined as $\lambda \sum_{j>k} T(g(x))_j$, where $T(v)$ sorts the elements of $v$ in descending order, and $\lambda \geq 0$ is a non-negative penalty. This regularization can encourage $g(x)$ to accumulate router probability mass per-input in the top $k$ elements, making the sampling process more deterministic per-input by the end of training. Note that this regularization doesn't affect the $k$-sparse training characteristics of the *MOESART* objective (2).

# 4 EXPERIMENTS

We study the performance of *MOESART* on recommender systems and image datasets in Section 4.1 and NLP tasks in 4.2. We include some ablation studies in Section 4.3.

## 4.1 EXPERIMENTS ON RECOMMENDER SYSTEMS AND IMAGE DATASETS

We study the performance of *MOESART* in recommender systems and image datasets. We compare with state-of-the-art $k$-sparse routers, including Top-$k$ (Shazeer et al., 2017), V-MoE (Ruiz et al., 2021), SMoE (Fedus et al., 2022; Zoph et al., 2022), Expert Choice Router (Zhou et al., 2022) and X-MoE (Chi et al., 2022). Note that Expert Choice Router obeys the $k$-sparsity on average across samples in a minibatch. We also include softmax router as a baseline.

**Datasets.** We consider multitask versions of two recommendation system datasets: MovieLens (Harper & Konstan, 2015) and Books (Ziegler et al., 2005). For MovieLens and Books, we have two tasks: classification task predicts whether user watches/reads a particular movie/book, regression problem predicts user's rating. For image datasets, we consider two multitask datasets: Multi-MNIST (Sabour et al., 2017) and Multi-FashionMNIST. There are two multi-class classification tasks (Hazimeh et al., 2021). Full details about each dataset are in Supplement Section S6.

**Experimental setup.** Although our exposition in Section 3 was for a single-task setting, the same router can be used in multi-task learning — multi-task requires multi-router MoE architecture (Ma

Table 1: Test loss, task-specific metrics and number of experts used per sample ($k/s$) while training for *MOESART* and existing purely sparse routing methods: (1) Top-$k$ (Shazeer et al., 2017), (2) V-MoE (Ruiz et al., 2021), (3) SMoE (Fedus et al., 2022), (4) Expert Choice Router (Zhou et al., 2022) and (5) X-MoE (Chi et al., 2022) across various datasets. Bold indicates statistical significance ($p$-value<0.05) over the best existing sparse router, using a one-sided unpaired t-test.

| Recommender Systems | | | | | |
|---|---|---|---|---|---|
| Dataset | Router | Test Loss ($\times 10^{-2}$) ↓ | Task-1 AUC ↑ | Task-2 MSE ↓ | Training $k/s$ ↓ |
| Books (TW=(0.1,0.9)) | Softmax (Dense) | $248.96 \pm 0.11$ | $55.31 \pm 0.08$ | $2.697 \pm 0.001$ | 9 |
| | Top-$k$ | $246.97 \pm 0.11$ | $56.64 \pm 0.08$ | $2.675 \pm 0.001$ | 4 |
| | V-MoE | $253.21 \pm 0.15$ | $55.11 \pm 0.06$ | $2.744 \pm 0.002$ | 4 |
| | SMoE | $249.88 \pm 0.13$ | $56.68 \pm 0.09$ | $2.707 \pm 0.001$ | 4 |
| | Expert Choice Router | $314.54 \pm 0.49$ | $56.22 \pm 0.10$ | $3.426 \pm 0.005$ | 4 |
| | X-MoE | $264.16 \pm 0.25$ | $56.18 \pm 0.10$ | $2.866 \pm 0.003$ | 4 |
| | *MOESART* | $\mathbf{242.47} \pm 0.09$ | $\mathbf{64.99} \pm 0.13$ | $\mathbf{2.626} \pm 0.001$ | 4 |
| Books (TW=(0.9,0.1)) | Softmax (Dense) | $74.69 \pm 0.04$ | $77.48 \pm 0.04$ | $2.697 \pm 0.002$ | 9 |
| | Top-$k$ | $74.63 \pm 0.03$ | $77.42 \pm 0.02$ | $2.683 \pm 0.002$ | 4 |
| | V-MoE | $75.96 \pm 0.04$ | $76.89 \pm 0.05$ | $2.768 \pm 0.003$ | 4 |
| | SMoE | $75.30 \pm 0.05$ | $77.13 \pm 0.05$ | $2.718 \pm 0.002$ | 4 |
| | Expert Choice Router | $82.83 \pm 0.06$ | $76.75 \pm 0.03$ | $3.403 \pm 0.009$ | 4 |
| | X-MoE | $78.25 \pm 0.05$ | $75.48 \pm 0.05$ | $2.890 \pm 0.003$ | 4 |
| | *MOESART* | $\mathbf{73.68} \pm 0.02$ | $\mathbf{78.03} \pm 0.03$ | $\mathbf{2.641} \pm 0.003$ | 4 |
| MovieLens (TW=(0.1,0.9)) | Softmax (Dense) | $73.96 \pm 0.02$ | $86.02 \pm 0.03$ | $0.7701 \pm 0.0002$ | 16 |
| | Top-$k$ | $77.72 \pm 0.06$ | $86.08 \pm 0.06$ | $0.8121 \pm 0.0007$ | 2 |
| | V-MoE | $75.67 \pm 0.04$ | $86.79 \pm 0.03$ | $0.7904 \pm 0.0005$ | 2 |
| | SMoE | $79.47 \pm 0.07$ | $85.36 \pm 0.06$ | $0.8303 \pm 0.0008$ | 2 |
| | Expert Choice Router | $81.14 \pm 0.04$ | $84.65 \pm 0.05$ | $0.8477 \pm 0.0005$ | 2 |
| | X-MoE | $77.59 \pm 0.12$ | $86.86 \pm 0.08$ | $0.8119 \pm 0.0013$ | 2 |
| | *MOESART* | $\mathbf{73.60} \pm 0.02$ | $\mathbf{87.33} \pm 0.03$ | $\mathbf{0.7684} \pm 0.0002$ | 2 |
| MovieLens (TW=(0.9,0.1)) | Softmax (Dense) | $41.93 \pm 0.02$ | $91.06 \pm 0.01$ | $0.7569 \pm 0.0002$ | 16 |
| | Top-$k$ | $41.90 \pm 0.02$ | $91.17 \pm 0.01$ | $0.7616 \pm 0.0004$ | 2 |
| | V-MoE | $42.11 \pm 0.03$ | $91.16 \pm 0.01$ | $0.7605 \pm 0.0006$ | 2 |
| | SMoE | $43.08 \pm 0.06$ | $90.86 \pm 0.03$ | $0.7917 \pm 0.0009$ | 2 |
| | Expert Choice Router | $44.94 \pm 0.04$ | $89.88 \pm 0.02$ | $0.8216 \pm 0.0006$ | 2 |
| | X-MoE | $44.66 \pm 0.04$ | $89.80 \pm 0.02$ | $0.7908 \pm 0.0006$ | 2 |
| | *MOESART* | $\mathbf{40.89} \pm 0.02$ | $\mathbf{91.61} \pm 0.01$ | $\mathbf{0.7430} \pm 0.0003$ | 2 |
| Image Tasks | | | | | |
| Dataset | Router | Test Loss ($\times 10^{-2}$) ↓ | Task-1 Accuracy ↑ | Task-2 Accuracy ↑ | Training $k/s$ ↓ |
| Multi-MNIST (TW=(0.5,0.5)) | Softmax (Dense) | $7.16 \pm 0.05$ | $98.16 \pm 0.02$ | $97.57 \pm 0.02$ | 8 |
| | Top-$k$ | $7.15 \pm 0.05$ | $98.12 \pm 0.02$ | $97.58 \pm 0.02$ | 4 |
| | V-MoE | $6.98 \pm 0.04$ | $98.16 \pm 0.02$ | $97.68 \pm 0.02$ | 4 |
| | SMoE | $6.90 \pm 0.05$ | $98.18 \pm 0.02$ | $97.69 \pm 0.02$ | 4 |
| | Expert Choice Router | $8.57 \pm 0.06$ | $97.80 \pm 0.02$ | $97.22 \pm 0.02$ | 4 |
| | X-MoE | $7.02 \pm 0.06$ | $98.21 \pm 0.02$ | $97.63 \pm 0.03$ | 4 |
| | *MOESART* | $\mathbf{5.86} \pm 0.03$ | $\mathbf{98.40} \pm 0.02$ | $\mathbf{97.92} \pm 0.02$ | 4 |
| Multi-FMNIST (TW=(0.5,0.5)) | Softmax (Dense) | $35.01 \pm 0.09$ | $88.10 \pm 0.05$ | $87.46 \pm 0.05$ | 5 |
| | Top-$k$ | $34.96 \pm 0.09$ | $88.06 \pm 0.05$ | $87.45 \pm 0.05$ | 2 |
| | V-MoE | $34.43 \pm 0.09$ | $88.04 \pm 0.05$ | $87.53 \pm 0.05$ | 2 |
| | SMoE | $34.68 \pm 0.09$ | $88.06 \pm 0.04$ | $87.52 \pm 0.06$ | 2 |
| | Expert Choice Router | $36.41 \pm 0.11$ | $87.50 \pm 0.08$ | $87.03 \pm 0.06$ | 2 |
| | X-MoE | $33.84 \pm 0.10$ | $88.05 \pm 0.08$ | $87.84 \pm 0.08$ | 2 |
| | *MOESART* | $\mathbf{32.85} \pm 0.11$ | $\mathbf{88.56} \pm 0.06$ | $\mathbf{88.02} \pm 0.07$ | 2 |

et al., 2018), where each task has a separate trainable router, but tasks have to select from a common set of experts. Total loss is the convex combination of loss for each task. For recommender systems, we train the network with a convex combination of the task-specific losses: binary cross-entropy (for classification) and mean squared error (for regression) with task weights (TW): $(\alpha, 1 - \alpha)$. We separately present results for two different task weight settings. For Multi-MNIST and Multi-FashionMNIST, we train with an equally weighted combination of cross-entropy losses. We used Adam optimizer, and we tuned the key hyperparameters using random grid search. After tuning, we train each model for 50 repetitions (using random initialization) and report the averaged test loss and task-specific metrics along with their standard errors. Full details about the respective MoE architectures and hyperparameter tuning for all routers are given in Supplement Section S6.

**Results.** In Table 1, we report the test loss, task-specific metrics and the number of experts for each input used during training across multiple recommender and image datasets. The results indicate that *MOESART* lead on all datasets with statistical significance, outperforming all state-of-the-art sparse routers e.g., Top-$k$, V-MoE, Expert Choice Router and X-MoE. Notably, *MOESART* can achieve 16% reduction in test loss on Multi-MNIST over all sparse routers. Similarly, we can observe that *MOESART* can achieve 15% (relative) improvement in ROC AUC over existing sparse routers on Books dataset, when the models are trained with $(0.1, 0.9)$ task weights combination.

## 4.2 DISTILLATION EXPERIMENTS ON NATURAL LANGUAGE PROCESSING TASKS

In this section, we study the performance of *MOESART* in the context of large language models (LLMs) on NLP tasks. In particular, we consider a setting where a pretrained LLM (non-MoE based) is distilled into an MoE based variant for more efficient inference while preserving or improving the performance. Zuo et al. (2022b) distilled BERT (Devlin et al., 2018) into its Sparse-MoE based MoEBERT. Specifically, the feedforward layers are replaced with MoE layers — this can result in a smaller number of (effective) parameters with per-input routing, thus allowing for more efficient inference. In the MoE layers of MoEBERT model, we consider Top-$k$ router (Shazeer et al., 2017) and our proposed *MOESART* router during distillation and evaluate the performance on the GLUE (Wang et al., 2019) and SQuAD benchmarks (Rajpurkar et al., 2016). We used $k = 2$ for both routers. More details about the distillation approach, datasets and tuning are summarized in Supplement Section S7. We adapted the codebase of Zuo et al. (2022b) written in HuggingFace.

**Results.** We report the performance metrics in Table 2 for 7 GLUE and 2 SQuAD benchmarks. *MOESART* consistently outperforms Top-$k$ on all GLUE and SQuAD benchmarks. On average, *MOESART* improves over Top-$k$ in prediction performance metrics by 0.5%.

Table 2: Performance metrics for distillation of BERT into MoEBERT with different routers on the GLUE and SQuAD development sets.

|  | GLUE | | | | | | | SQuAD | |
|---|---|---|---|---|---|---|---|---|---|
|  | **RTE** | **CoLA** | **MRPC** | **SST-2** | **QNLI** | **QQP** | **MNLI** | **v1.1** | **v2.0** |
|  | Acc ↑ | Mcc ↑ | F1 ↑ | Acc ↑ | Acc ↑ | F1 ↑ | m/mm ↑ | F1 ↑ | F1 ↑ |
| MoEBERT with Top-$k$ | 68.95 | 57.86 | 86.76 | 92.78 | 91.63 | 88.10 | 85.14 | 88.38 | 78.73 |
| MoEBERT with *MOESART* | **70.40** | **58.20** | **88.24** | **93.12** | **91.95** | **88.27** | **85.21** | **88.43** | **79.02** |

## 4.3 ABLATION STUDIES FOR *MOESART*

We perform multiple ablation studies to show different elements of *MOESART*: (i) Effect of sampling strategies during training on out-of-sample generalization on SVHN. (ii) Effect of varying $k$ on model's performance on SVHN. (iii) Spatial structure of learnt embeddings on MovieLens. Additional ablation studies studying (a-b) bias and performance of different adjustment strategies, (c) effect of trimmed lasso regularization on performance, and (d) performance under additional load balancing requirements for *MOESART* are included in Supplement Section S3.

**Effect of sampling strategy.** We first study the effect of different sampling strategies on performance of Sparse MoE. We consider two options: (a) sampling with replacement (b) sampling without replacement. For this ablation study, we consider *MOESART* with $k = 2$ on SVHN dataset. We show the evolution of out-of-sample loss during the course of training in Figure 3. We can observe that sampling without replacement appears to be more stable and achieves better out-of-sample loss. We hypothesize that the sub-optimal performance of sampling with replacement can be attributed to

two reasons: (a) The variance is smaller for sampling without replacement, (b) For sampling with replacement, with $k = 2$, if the same expert gets sampled twice for an input, this example is unable to contribute to the gradient update of the router parameters.

**Effect of varying $k$.** Here, we study the effect of $k$ on MoE performance. Note that number of experts is fixed to $8$. We train Sparse MoE models for different values of $k = \{2, 4, 6\}$ and visualize the generalization performance in Figure 4. For comparison, we visualize the results for both Top-$k$ and *MOESART*. We also show the performance of softmax router. We observe that both sparse routers improve in performance as $k$ is increased. Notably, we consistently see a significant gap in performance between Top-$k$ and *MOESART* for each $k$ setting.

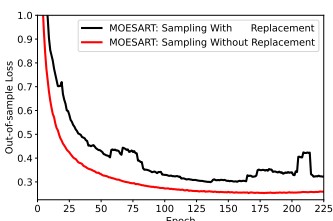

Figure 3: Out-of-sample loss for *MOESART* for different sampling strategies on SVHN.

**Spatial structure of learnt embeddings.** We consider Movie-Lens dataset and the same multi-task MoE-based architecture as detailed in Section S6.2 — however, we use 4 experts for this exercise. The architecture has user and movie learnable embedding layers, which are concatenated and fed into an MoE layer with 4 experts, followed by a task-specific head for classification task and regression task. We set $k = 2$ for Top-$k$ and *MOE-SART*. We optimize with Adam with $5 \times 10^{-5}$ learning rate with a 512 batch size. We visualize the embeddings learnt by Top-$k$ and *MOESART* routers in Figure 5. We use Uniform Manifold Approximation and Projection (UMAP) (McInnes et al., 2018) to project the concatenated embeddings of each input to a two-dimensional space. Each data point represents an input to be routed. Each color stands for the top expert that each input is assigned to. *MOESART* seems to provide better specialization of experts, where the distinct distribution handled by each expert is much clearer (5a,5b) in comparison to that for Top-$k$ router (5c,5d). The learnt embeddings are more disentangled for *MOESART* in comparison to Top-$k$.

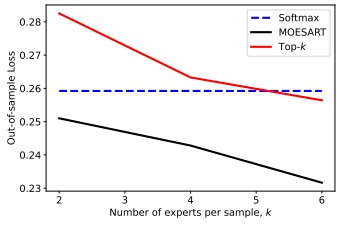

Figure 4: Out-of-sample loss achieved by *MOESART* and Top-$k$ for different $k$ on SVHN.

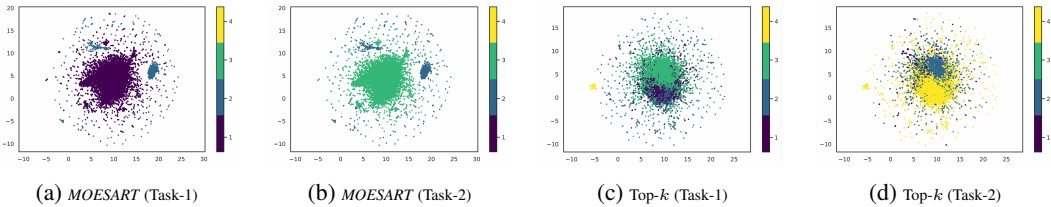

| (a) *MOESART* (Task-1) | (b) *MOESART* (Task-2) | (c) Top-$k$ (Task-1) | (d) Top-$k$ (Task-2) |

Figure 5: UMAP projection of user/movie embeddings learnt by the different routers on (multi-task) MovieLens. Each data point represents an embedding input to be routed. Color denotes expert index. *MOESART* seems to provide better specialization of experts, where the distinct distribution handled by each expert is much clearer (5a,5b) in comparison to that for Top-$k$ router (5c,5d).

## 5 CONCLUSION

We proposed a sampling-based routing mechanism with *MOESART*. Our approach aims to learn a sparse approximation of the softmax based classical MoE. We achieve this through sampling and novel expert reweighting strategies. The sparse approximation learnt by our approach appears to be substantially better than that learnt by Top-$k$ router and its variants. *MOESART* allows conditional training as it is $k$-sparse during training similar to Top-$k$ style routing strategies. We performed large-scale experiments on 14 datasets from various domains. On standard vision and recommender systems, *MOESART* achieves up to $16\%$ (relative) smaller out-of-sample loss and up to $15\%$ (relative) improvement in ROC AUC over state-of-the-art $k$-sparse routers, e.g., Top-k, V-MoE, Expert Choice Router and X-MoE. Moreover, for distillation in NLP tasks, *MOESART* router can consistently outperform Top-$k$ router in MoEBERT model on GLUE and SQuAD benchmarks.

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

SUPPLEMENTARY MATERIAL.

## S1 PROOF FOR $\tilde{g}_i$ IN (III) BEING AN UNBIASED ESTIMATOR OF $g_i$

In this section, we study the biasness of adjustment strategy ((iii) in the main paper), as given by

$$\tilde{o}_i := \begin{cases} o_i + \log(r_i) - \log(kg_i) & i \in s, \\ -\infty & i \notin s, \end{cases} \tag{S1}$$

Recall that $\tilde{g}_i = \exp(\tilde{o}_i)/\sum_{l\in[n]}\exp(\tilde{o}_l)$. We present the following theorem:

**Theorem 1** *For each $i \in [n]$, if sampling (with replacement) is performed from the softmax probability, $g_i \propto exp(o_i)$, then $\tilde{g}_i$ (corresponding to $\tilde{o}_i$ defined in (S1) is an unbiased estimator of $g_i$, i.e., $\mathbb{E}[\tilde{g}_i] = g_i$.*

Before, we can show the proof for Theorem 1, we present the following result:

**Lemma 1** *For $\tilde{o}_i$ defined as in (S1), the following equality holds:*

$$\sum_{j\in[n]} exp(\tilde{o}_j) = \sum_{l\in[n]} exp(o_l). \tag{S2}$$

Proof for Lemma 1:

$$\sum_{j\in[n]} \exp(\tilde{o}_j) = \sum_{j\in s}\exp(\tilde{o}_j) + \sum_{j\notin s}\exp(\tilde{o}_j) = \sum_{j\in s}\exp(\tilde{o}_j) \tag{S3}$$

$$= \sum_{j\in s}\exp(o_j - \log(kg_j) + \log(r_j)) = \sum_{j\in s}\frac{r_j\exp(o_j)}{kg_j} \tag{S4}$$

$$= \sum_{j\in s}\frac{r_j\exp(o_j)}{k}\frac{\sum_{l=1}^{n}\exp(o_l)}{\exp(o_j)} \tag{S5}$$

$$= \frac{1}{k}\left(\sum_{j\in s}r_j\right)\left(\sum_{l=1}^{n}\exp(o_l)\right) = \frac{1}{k}(k)\left(\sum_{l=1}^{n}\exp(o_l)\right) \tag{S6}$$

$$= \sum_{l=1}^{n}\exp(o_l). \quad\blacksquare \tag{S7}$$

Proof for Theorem 1:

$$\mathbb{E}_r\left[\tilde{g}_j\right] = \mathbb{E}_r\left[\frac{\exp(\tilde{o}_i)}{\sum_{l\in[n]}\exp(\tilde{o}_l)}\right] \tag{S8}$$

$$= \mathbb{E}_r\left[\frac{\exp(\tilde{o}_i)}{\sum_{l=1}^{n}\exp(o_l)}\right] \quad \text{using Lemma 1} \tag{S9}$$

$$= \mathbb{E}_r\left[\frac{\exp(o_i - \log(kg_i) + \log(r_i))}{\sum_{l=1}^{n}\exp(o_l)}\right] \tag{S10}$$

$$= \frac{1}{k}\mathbb{E}_r\left[\frac{r_i}{g_i}\frac{\exp(o_i)}{\sum_{l=1}^{n}\exp(o_l)}\right] \tag{S11}$$

$$= \frac{1}{k}\mathbb{E}_r[r_i] = \frac{1}{k}(kg_i) = g_i. \quad\blacksquare \tag{S12}$$

## S2 VECTORIZED MINI-BATCH SAMPLING FOR *MOESART*

---

**Algorithm 1** Vectorized mini-batched sampling with/without replacement

---

**Input**: $G \in \mathbb{R}^{B,n}$ — Expert probability mass function $g(x)$ for each sample $x$ in batch $\mathcal{B}$ in matrix form.
**Output**: $R \in \mathbb{Z}_{\geq 0}^{B,n}$ — Count Matrix, which records number of times each expert is sampled for each input in the minibatch. $R$ is used to get the indices set $S$.
**Hyperparameters**: $k$, replace.

1: Denote sample indices in mini-batch, $\mathcal{B} = \{1, \ldots, B\}$
2: Initialize count matrix, $R = 0 \in \mathbb{Z}_{\geq 0}^{B,n}$.
3: **for** $\ell = 1, \ldots, k$ **do**
4:     Initialize $R_l = 0 \in \mathbb{Z}_{\geq 0}^{B,n}$.
5:     Sample $\mathbf{u} \in \mathbb{R}^B \sim \mathcal{U}(0, 1)$.
6:     Compute: $C[b, i] = \sum_{q=1}^{i} G[b, q] \;\; \forall i \in [n], \;\; \forall b \in \mathcal{B}$.
7:     Define $s$ such that $s[b] = \arg\max_{i \in [n]} 1\{u[b] < C[b, i]\} \;\; \forall b \in \mathcal{B}$.
8:     Update $R_l[b, s[b]] = 1 \;\; \forall b \in \mathcal{B}$.
9:     Find indices $\mathcal{Z} \subseteq \mathcal{B}$ such that $\sum_i G[z, i] = 0 \;\; \forall z \in \mathcal{Z}$.
10:     Update $R_l[z, s[z]] = 0 \;\; \forall z \in \mathcal{Z}$
11:     Update $R \leftarrow R + R_l$.
12:     **if** not replace **then**
13:         Update $G[b, s[b]] = 0 \;\; \forall b \in \mathcal{B}$.
14:         Row-sum normalize $G \leftarrow G \oslash (G \cdot \mathbf{1}_n \cdot \mathbf{1}_n^T)$.
15:     **end if**
16: **end for**

---

## S3 ADDITIONAL ABLATION STUDIES

In this section, we show four ablation studies:

(a) Bias of different adjustment strategies in (i)-(ii) and proposed strategy in (3).
(b) Performance comparison of *MOESART* with different adjustment strategies in (i)-(ii) and (3) for training of Sparse-MoE on Image datasets.
(c) Effect of trimmed lasso regularization on performance of *MOESART* on Recommender Systems.
(d) Performance of *MOESART* under additional load balancing requirements.

### S3.1 BIAS OF DIFFERENT ADJUSTMENT STRATEGIES IN (I)-(II) AND OUR PROPOSED STRATEGY IN (3)

In this ablation study, we empirically study the bias of different adjustment strategies in (i)-(ii) and the proposed strategy in (3). For this study, we consider different choices for $g$: uniform, random, decaying. We measure the bias with the metric $\|\mathbb{E}[\tilde{g}] - g\|_2$. To be precise, this metric $\|\mathbb{E}[\tilde{g}] - g\|_2$ is defined as:

$$\|(\mathbb{E}[\tilde{g}_1], \cdots, \mathbb{E}[\tilde{g}_n]) - (g_1, \cdots, g_n)\|_2 \qquad (S13)$$

We show the metric in Fig. S1 for different distribution shapes for $g$. For uniform setting, we observe the bias to be very similar across different adjustment strategies. However, for sufficiently non-uniform distributions, we can observe that there can be a

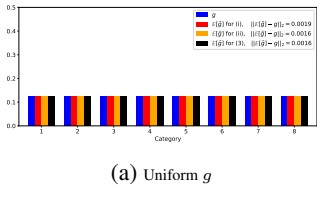

(a) Uniform $g$

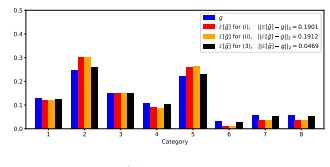

(b) Random $g$

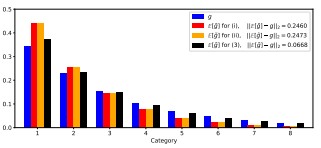

(c) Exponentially Decaying $g$

Figure S1: Comparison of bias of different adjustment strategies (i)-(iii) and the proposed strategy in (3) for different $g$.

significant gap in the bias for proposed strategy in (3) and the adjustment strategies in (i)-(ii). In Sparse MoE, the distribution $g(x)$ can be very different across different inputs, hence our proposed strategy is expected to have a lower bias overall across samples. A smaller bias can lead to improved learning. We observe this to be the case in Sparse MoE, which we show in the next ablation study.

## S3.2 PERFORMANCE COMPARISON OF OUR PROPOSED ADJUSTMENT IN (3) VERSUS (I)-(II) ON IMAGE DATASETS.

In this ablation study, we compare the performance of *MOESART* under different adjustment strategies for computing $\tilde{g}$. In particular, we compare the proposed strategy in (3) without replacement against the strategies in (i)-(ii) without replacement. Note that (i) and (ii) are equivalent when sampling without replacement.

We show results for different adjustment strategies in Table S1 on multi-task image datasets. We also include Top-$k$ for comparison. We can see our proposed strategy in (3) can be significantly better than the strategies in (i)-(ii). We can also observe that both sampling-based strategies (i-ii) and (3) significantly outperform Top-$k$ routing. This potentially highlights that Top-$k$ greedy routing has a large selection bias, which is reduced by sampling based approaches.

Table S1: Comparison of test loss and task-specific metrics for *MOESART* with the proposed adjustment strategy in (3) against other strategies in (i)-(ii). We also include Top-$k$ for reference.

| Image Tasks | | | | | |
|---|---|---|---|---|---|
| Dataset | Model | Test Loss ($\times 10^{-2}$) ↓ | Task-1 Accuracy ↑ | Task-2 Accuracy ↑ | Training $k/s$ ↓ |
| Multi-MNIST (TW=(0.5,0.5)) | Top-$k$ | $7.15 \pm 0.05$ | $98.12 \pm 0.02$ | $97.58 \pm 0.02$ | 4 |
| | *MOESART* with (i),(ii) | $6.15 \pm 0.05$ | $98.33 \pm 0.02$ | $\mathbf{97.92} \pm 0.02$ | 4 |
| | *MOESART* with (3) | $\mathbf{5.86} \pm 0.03$ | $\mathbf{98.40} \pm 0.02$ | $97.92 \pm 0.02$ | 4 |
| Multi-FMNIST (TW=(0.5,0.5)) | Top-$k$ | $34.96 \pm 0.09$ | $88.06 \pm 0.05$ | $87.45 \pm 0.05$ | 2 |
| | *MOESART* with (i),(ii) | $34.15 \pm 0.09$ | $88.19 \pm 0.05$ | $87.71 \pm 0.05$ | 2 |
| | *MOESART* with (3) | $\mathbf{32.85} \pm 0.11$ | $\mathbf{88.56} \pm 0.06$ | $\mathbf{88.02} \pm 0.07$ | 2 |

## S3.3 EFFECT OF TRIMMED LASSO REGULARIZATION ON PERFORMANCE ON RECOMMENDER SYSTEMS

In this section, we perform ablation study for the trimmed lasso regularization on recommender systems datasets i.e., MovieLens and Books. We had performed a large set of 500 tuning trials, which tuned over multiple hyperparameters including the trimmed lasso regularization $\lambda$ in the set $\{0, 0.0001, 0.001, 0.01, 0.1, 1.0, 10.0\}$. We identify the best tuning trial for both cases $\lambda = 0$ and $\lambda > 0$ based on the validation set and report the test performance for the two cases in Table S2. We can observe that for both datasets, trimmed lasso regularization can provide some performance gain over the case without trimmed lasso regularization.

Table S2: Comparison of test loss for best trial for *MOESART* without and with Trimmed Lasso Regularization on Recommender system datasets.

| Dataset | Model | Test Loss ($\times 10^{-2}$) ↓ |
|---|---|---|
| MovieLens | *MOESART* without Trimmed Lasso | 73.85 |
| | *MOESART* with Trimmed Lasso | $\mathbf{73.40}$ |
| Books | *MOESART* without Trimmed Lasso | 246.50 |
| | *MOESART* with Trimmed Lasso | $\mathbf{243.02}$ |

## S3.4 PERFORMANCE UNDER LOAD BALANCING

Load balancing is also another important consideration for efficiency in Sparse MoE models. Load balancing requires similar number of examples to be routed to each expert. Traditionally, an auxiliary loss is added explicity to achieve load balancing. Shazeer et al. (2017) and Fedus et al. (2022) proposed two different auxiliary losses to achieve load balancing. Many follow-up works on Top-$k$ based routing (Zoph et al., 2022; Chi et al., 2022; Xie et al., 2023) impose one of these auxiliary losses. We add an auxiliary loss as the one imposed by Fedus et al. (2022) to Top-$k$ router and *MOESART* router and compare the performance of these methods when the load balancing regularization is designed to achieve $99\%$ load balancing. We also compare against the Expert Choice

Router by Zhou et al. (2022), which is designed to achieve $100\%$ load balancing across experts. We again consider (multi-task) MovieLens dataset. *MOESART* with load balancing achieves the best performance as shown in Table S3.

Table S3: Test loss with load balancing on MovieLens.

| Model | Test Loss ($\times 10^{-2}$) $\downarrow$ |
|---|---|
| Top-$k$ | $77.72 \pm 0.03$ |
| Expert Choice Router | $81.14 \pm 0.04$ |
| *MOESART* | $\mathbf{74.04} \pm 0.03$ |

## S4 COMPARISON WITH DIFFERENTIABLE ROUTERS

In this section, we compare our routing approach with some state-of-the-art differentiable routers. In particular, we compare with two routers: (i) DSelect-k (Hazimeh et al., 2021), (ii) COMET (Ibrahim et al., 2023). We tuned these differentiable routers for their respective hyperparameters with random search over 500 tuning trials and report the averages across 50 runs for their best hyperparameters. We compare across various datasets and report the performance metrics in Table S4.

Table S4: Comparison of test loss and task-specific metrics for *MOESART* against differentiable routing methods: (1) DSelect-$k$ (Hazimeh et al., 2021), (2) COMET (Ibrahim et al., 2023), across various datasets. Note that these differentiable routers can only support conditional training partially with customized implementations, hence they are more expensive during training.

| Recommender Systems | | | | |
|---|---|---|---|---|
| Dataset | Model | Test Loss ($\times 10^{-2}$) $\downarrow$ | Task-1 AUC $\uparrow$ | Task-2 MSE $\downarrow$ |
| Books (TW=(0.1,0.9)) | Softmax (Dense) | $248.96 \pm 0.11$ | $55.31 \pm 0.08$ | $2.697 \pm 0.001$ |
| | DSelect-$k$ | $241.64 \pm 0.14$ | $59.05 \pm 0.14$ | $2.615 \pm 0.002$ |
| | COMET | $\mathbf{241.13} \pm 0.11$ | $\mathbf{66.13} \pm 0.09$ | $\mathbf{2.612} \pm 0.001$ |
| | *MOESART* | $242.47 \pm 0.09$ | $64.99 \pm 0.13$ | $2.626 \pm 0.001$ |
| Books (TW=(0.9,0.1)) | Softmax (Dense) | $74.69 \pm 0.04$ | $77.48 \pm 0.04$ | $2.697 \pm 0.002$ |
| | DSelect-$k$ | $74.22 \pm 0.03$ | $77.61 \pm 0.02$ | $\mathbf{2.632} \pm 0.002$ |
| | COMET | $74.29 \pm 0.03$ | $77.30 \pm 0.02$ | $2.636 \pm 0.002$ |
| | *MOESART* | $\mathbf{73.68} \pm 0.02$ | $\mathbf{78.03} \pm 0.03$ | $2.641 \pm 0.003$ |
| MovieLens (TW=(0.1,0.9)) | Softmax (Dense) | $73.96 \pm 0.02$ | $86.02 \pm 0.03$ | $0.7701 \pm 0.0002$ |
| | DSelect-$k$ | $73.81 \pm 0.03$ | $\mathbf{87.79} \pm 0.03$ | $0.7715 \pm 0.0003$ |
| | COMET | $74.19 \pm 0.03$ | $87.67 \pm 0.07$ | $0.7756 \pm 0.0004$ |
| | *MOESART* | $\mathbf{73.60} \pm 0.02$ | $87.33 \pm 0.03$ | $\mathbf{0.7684} \pm 0.0002$ |
| MovieLens (TW=(0.9,0.1)) | Softmax (Dense) | $41.93 \pm 0.02$ | $91.06 \pm 0.01$ | $0.7569 \pm 0.0002$ |
| | DSelect-$k$ | $41.40 \pm 0.03$ | $91.44 \pm 0.01$ | $0.7582 \pm 0.0005$ |
| | COMET | $41.25 \pm 0.04$ | $91.45 \pm 0.02$ | $0.7563 \pm 0.0008$ |
| | *MOESART* | $\mathbf{40.89} \pm 0.02$ | $\mathbf{91.61} \pm 0.01$ | $\mathbf{0.7430} \pm 0.0003$ |
| Image Tasks | | | | |
| Dataset | Model | Test Loss ($\times 10^{-2}$) $\downarrow$ | Task-1 Accuracy $\uparrow$ | Task-2 Accuracy $\uparrow$ |
| Multi-MNIST (TW=(0.5,0.5)) | Softmax (Dense) | $7.16 \pm 0.05$ | $98.16 \pm 0.02$ | $97.57 \pm 0.02$ |
| | DSelect-$k$ | $6.93 \pm 0.06$ | $98.14 \pm 0.02$ | $97.68 \pm 0.03$ |
| | COMET | $6.83 \pm 0.06$ | $98.21 \pm 0.02$ | $97.63 \pm 0.03$ |
| | *MOESART* | $\mathbf{5.86} \pm 0.03$ | $\mathbf{98.40} \pm 0.02$ | $\mathbf{97.92} \pm 0.02$ |
| Multi-FMNIST (TW=(0.5,0.5)) | Softmax (Dense) | $35.01 \pm 0.09$ | $88.10 \pm 0.05$ | $87.46 \pm 0.05$ |
| | DSelect-$k$ | $36.88 \pm 0.21$ | $87.37 \pm 0.07$ | $86.61 \pm 0.09$ |
| | COMET | $34.88 \pm 0.13$ | $87.97 \pm 0.06$ | $87.42 \pm 0.06$ |
| | *MOESART* | $\mathbf{32.85} \pm 0.11$ | $\mathbf{88.56} \pm 0.06$ | $\mathbf{88.02} \pm 0.07$ |

*MOESART* appears to be quite competitive with differentiable routers. Surprisingly, *MOESART* can sometimes even outperform differentiable routers. We hypothesize that this maybe due to the choice of parameterization. These routers rely on a particular activation function (Smooth-Step function (Hazimeh et al., 2020)) to achieve binary state for sparse inference. The binary gates snap into place across samples reaching a permanent state of $0$ or $1$ as training progresses. This is understandably a desired property of Smooth-Step activation for conditional computation, however it means that the model cannot update decisions regarding the choice of expert once this permanent state is reached. In contrast, our sampling-based approach can allow for exploration throughout training, which can be beneficial.

We would like to remind the reader that although these routers can allow conditional inference, they are less appealing in terms of conditional training. These routers are dense-to-sparse routers, and hence can only partially allow for conditional training (during a later stage of training with customized implementations). Hence, these routers are more expensive as during the dense phase of training they require computing gradients with respect to a larger set of experts. In contrast, our proposed routing approach *MOESART* completely allows conditional training and conditional inference.

## S5  SENSITIVITY OF *MOESART* TO HYPERPARAMETER TUNING

Here, we study the sensitivity of *MOESART* to hyperparameter tuning and show that router can be beneficial in terms of hyperparameter tuning over Top-$k$ style routers. We perform a large set of tuning trials and perform a bootstrapping procedure to see whether *MOESART* helps in reducing the hyperparameter tuning overload. We first describe the bootstrapping procedure.

**Bootstrapping procedure for studying hyperparameter tuning**  We performed 500 tuning trials for each router optimizing over various different hyperparameters and performed a bootstrapping procedure as outlined below:

- Randomly sample $s$ ($s \in \{1, 2, 5, 10, 15, \cdots, 250\}$) trials from the bag of a larger set of 500 trials.
- Find the trial with the best validation loss.
- Compute the test loss for that trial.
- Repeat this exercise for 1000 times.
- Compute the average test loss across the best selected trials.

We visualize the impact of number of trials on performance in Figure S2. *MOESART* can achieve the same level of performance as Top-$k$ router with much lesser number of hyperparameter trials. This indicates that *MOESART* is not too heavily dependent on a very restricted set of hyperparameter values. We visualize this for various datasets in Fig. S2. We see tuning reduction by a factor of $100\times$ for *MOESART* over Top-$k$. Alternatively, even a small number of trials e.g., 2-10 can show a significant gain over Top-$k$ router.

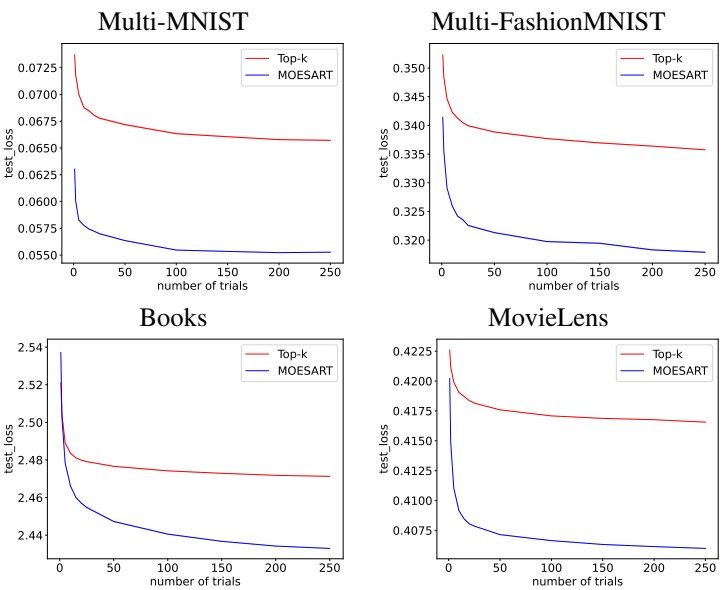

Figure S2: *Sensitivity of MOESART to hyperparameter tuning. MOESART* can achieve the same level of performance as Top-$k$ with significantly lesser number of hyperparameter trials. We see tuning reduction by $100\times$ for *MOESART* over Top-k.

## S6 ADDITIONAL DETAILS FOR SECTION 4.1

### S6.1 DATASETS

**SVHN.** We consider Street View Housing Numbers (SVHN) (Netzer et al., 2011) dataset. SVHN is a dataset of $\sim 600,000$ images obtained from house numbers in Google Street View images. We used the original training (#train: 73257) and test (#test: 26032) splits. We subsampled 50,000 samples from the extra split as validation dataset.

**MovieLens.** MovieLens (Harper & Konstan, 2015) is a movie recommendation dataset containing records for $\sim 4,000$ movies and $\sim 6,000$ users. Following Wang et al. (2020), for every user-movie pair, we construct two tasks. Task 1 is a binary classification problem for predicting whether the user will watch a particular movie. Task 2 is a regression problem to predict the user's rating (in $\{1, 2, \cdots, 5\}$) for a given movie. We use 1.6 million samples for training and $200,000$ for each of the validation and testing sets.

**Books.** Books (Ziegler et al., 2005) is a book recommendation dataset containing records for $\sim 105k$ users and $\sim 340k$ books. We filter users and books with each atleast 5 records. This gives a subset of $18,960$ users and $31,070$ books. This gives a subset of $556,724$ records. Similar to MovieLens above, for every user-book pair, we construct two tasks. Task 1 is a binary classification problem for predicting whether the user will read a particular book. Task 2 is a regression problem to predict the user's rating (in $\{1, 2, \cdots, 10\}$) for a given book. We use 389,706 samples for training and 83,509 for each of the validation and testing sets.

**Multi-MNIST and Multi-FashionMNIST.** We consider multi-task variants of MNIST and FashionMNIST (Deng, 2012). The datasets are constructed in a similar fashion as given in Sabour et al. (2017); Hazimeh et al. (2021): (i) uniformly sample two images from MNIST and overlay them on top of each other, and (ii) shift one digit towards the top-left corner and the other digit towards the bottom-right corner (by 4 pixels in each direction). This procedure leads to $36 \times 36$ images with some overlap between the digits. We consider two classification tasks: Task 1 is to classify the top-left item and Task 2 is to classify the bottom-right item. We use 100,000 samples for training, and $20,000$ samples for each of the validation and testing sets.

### S6.2 ARCHITECTURES

**SVHN.** We use an architecture with an MoE-layer followed by a stack of 3 dense layers: the first two have 50 ReLU-activated units and the third has 10 units followed by a softmax. The MoE layer consists of 8 experts, each of which is a CNN that is composed (in order) of: (i) convolutional layer 1 (kernel size = 5, #filters = 10, ReLU-activated) followed by max pooling, (ii) convolutional layer 2 (kernel size=5, #filters = 20, ReLU-activated) followed by max pooling, and (iii) a sequence of 2 ReLU-activated dense layers with 50 units each.

**Multi-MNIST and Multi-FashionMNIST.** We use a multi-router MoE with 8 and 5 experts for Multi-MNIST and Multi-FashionMNIST respectively. Each of the experts is a CNN that is composed (in order) of: (i) convolutional layer 1 (kernel size = 5, #filters = 10, ReLU-activated) followed by max pooling, (ii) convolutional layer 2 (kernel size=5, #filters = 20, ReLU-activated) followed by max pooling, and (iii) a sequence of 2 ReLU-activated dense layers with 50 units each. The subnetwork specific to each of the 2 tasks is composed of a stack of 3 dense layers: the first two have 50 ReLU-activated units and the third has 10 units followed by a softmax.

**MovieLens.** We consider a multi-router MoE architecture, where each task is associated with a separate router. The MoE architecture consists of a shared bottom subnetwork comprising two embedding layers (for users and movies). The 128-dimensional embeddings from both layers are concatenated and fed into an MoE Layer of 16 experts, where each expert is a ReLU-activated dense layer with 256 units, followed by a dropout layer (with a dropout rate of 0.5). For each of the two tasks, the corresponding convex combination of the experts is fed into a task-specific subnetwork. The subnetwork is composed of a dense layer (ReLU-activated with 256 units) followed by a single unit that generates the final output of the task.

**Books.**    We consider a multi-router MoE architecture, where each task is associated with a separate router. The MoE architecture consists of a shared bottom subnetwork comprising two embedding layers (for users and books/jokes). The 64-dimensional embeddings from both layers are concatenated and fed into an MoE Layer of 9 experts, where each expert is a ReLU-activated dense layer with 128 units, followed by a dropout layer (with a dropout rate of 0.5). For each of the two tasks, the corresponding convex combination of the experts is fed into a task-specific subnetwork. The subnetwork is composed of a dense layer (ReLU-activated with 256 units) followed by a single unit that generates the final output of the task.

### S6.3    Hyperparameters and Tuning

We performed 500 tuning trials for each router with a random search over the hyperparameter space described below (for each dataset). For each router, we tune the optimization and router-specific hyperparameters and use the validation loss as the tuning metric. After tuning, we train each model for 50 repetitions (using random initializations) and report the averaged results along with the standard errors in Table 1.

**SVHN.**

- Learning Rates: $1 \times 10^{-4}$ for Adam.
- Batch-size: 512.
- Epochs: 250 with early stopping (patience=50) based on validation set.
- Trimmed Lasso, $\lambda$: 0.0 for *MOESART*.
- $n$ (number of experts): 8.
- $k$: 2 for all sparse (trainable) routers.
- Number of tuning trials per router: 1

**MovieLens.**

- Learning Rates: Uniform in the log range $[5 \times 10^{-5}, 5 \times 10^{-4}]$ for Adam.
- Batch-size: 512.
- Epochs: 100 with early stopping (patience=25) based on validation set.
- Trimmed Lasso, $\lambda$: Discrete uniform in the set $\{0.0, 0.01, 0.1, 1, 10\}$ for *MOESART*.
- $n$ (number of experts): 16.
- $k$: 2 for all sparse (trainable) routers.
- Number of tuning trials per router: 500

**Books.**

- Learning Rates: Uniform in the log range $[5 \times 10^{-5}, 5 \times 10^{-4}]$ for Adam.
- Batch-size: 2048.
- Epochs: 100 with early stopping (patience=25) based on validation set.
- Trimmed Lasso, $\lambda$: Discrete uniform in the set $\{0.0, 0.01, 0.1, 1, 10\}$ for *MOESART*.
- $n$ (number of experts): 9.
- $k$: 4 for all sparse (trainable) routers.
- Number of tuning trials per router: 500

**Multi-MNIST.**

- Learning Rates: Uniform in the log range $[1 \times 10^{-4}, 1 \times 10^{-3}]$ for Adam.
- Batch-size: 512.
- Epochs: 200 with early stopping (patience=25) based on validation set.
- Trimmed Lasso, $\lambda$: Discrete uniform in the set $\{0.0, 0.01, 0.1, 1, 10\}$ for *MOESART*.
- $n$ (number of experts): 8.
- $k$: 4 for all sparse (trainable) routers.
- Number of tuning trials per router: 500

**Multi-FashionMNIST.**

- Learning Rates: Uniform in the log range $[1 \times 10^{-4}, 1 \times 10^{-3}]$ for Adam.

- Batch-size: 512.
- Epochs: 200 with early stopping (patience=25) based on validation set.
- Trimmed Lasso, $\lambda$: Discrete uniform in the set $\{0.0, 0.01, 0.1, 1, 10\}$ for *MOESART*.
- $n$ (number of experts): 5.
- $k$: 2 for all sparse (trainable) routers.
- Number of tuning trials per router: 500

## S7 ADDITIONAL DETAILS FOR SECTION 4.2

### S7.1 DATASETS

**GLUE.** General Language Understanding Evaluation (GLUE) benchmark (Wang et al., 2019), is a collection of natural language understanding tasks. Following previous works on model distillation, we consider SST-2 (Socher et al., 2013), CoLA (Warstadt et al., 2019), MRPC (Dolan & Brockett, 2005), STSB (Cer et al., 2017), QQP, and MNLI (Williams et al., 2018) and exclude STS-B (Cer et al., 2017) and WNLI (Levesque et al., 2012) in the experiments. The datasets are briefly summarized below:

- SST-2 (Socher et al., 2013) is a binary single-sentence classification task that classifies movie reviews to positive or negative;
- CoLA (Warstadt et al., 2019) is a linguistic acceptability task;
- MRPC (Dolan & Brockett, 2005) is a paraphrase detection task;
- QQP is a duplication detection task;
- MNLI (Williams et al., 2018), QNLI (Rajpurkar et al., 2016), and RTE (Dagan et al., 2006) are natural language inference tasks.

Dataset details are summarized in Table S5.

Table S5: Summary of GLUE benchmark.

| Corpus | Task | #Train | #Dev | #Test | #Label | Metrics |
|---|---|---|---|---|---|---|
| Single-Sentence Classification (GLUE) | | | | | | |
| CoLA | Acceptability | 8.5k | 1k | 1k | 2 | Matthews correlation |
| SST-2 | Sentiment | 67k | 872 | 1.8k | 2 | Accuracy |
| Pairwise Text Classification (GLUE) | | | | | | |
| MNLI | NLI | 393k | 20k | 20k | 3 | Accuracy |
| RTE | NLI | 2.5k | 276 | 3k | 2 | Accuracy |
| QQP | Paraphrase | 364k | 40k | 391k | 2 | Accuracy/F1 |
| MRPC | Paraphrase | 3.7k | 408 | 1.7k | 2 | Accuracy/F1 |
| QNLI | QA/NLI | 108k | 5.7k | 5.7k | 2 | Accuracy |

**SQuAD.** We evaluate our sparse routing approaches on SQuAD benchmarks (Rajpurkar et al., 2016; 2018). This task is treated as a sequence labeling problem, where we predict the probability of each token being the start and end of the answer span. Statistics of the question answering datasets are summarized in Table S6.

Table S6: Summary of SQuAD benchmark.

| Corpus | Task | #Train | #Dev | Metrics |
|---|---|---|---|---|
| SQuAD v1.1 | Question Answering | 87.6k | 10.6k | F1/EM |
| SQuAD v2.0 | Question Answering | 130k | 11.9k | F1/EM |

### S7.2 TUNING PROCEDURE FOR MOEBERT

Following Zuo et al. (2022b), we followed the 3-step process as outlined in the MoEBERT code-base[1]:

---

[1] https://github.com/SimiaoZuo/MoEBERT

- Finetune BERT on downstream task. We used finetuned BERT from HuggingFace on each downstream task.

- Compute importance weights in FFN layers to construct an MoEBERT model, where FFN layers are replaced with MoE layers with the weight assignment strategy in (Zuo et al., 2022b).

- Distill BERT into MoEBERT on the downstream task with a combination of cross-entropy loss and layer-wise discrepancy loss. For MoEBERT with Top-$k$ and *MOESART*, we performed a tuning procedure and picked the best results based on development datasets. We performed a grid search over the following sets of hyperparameters:

  - Learning Rate: We used same learning rates as the optimal ones reported for each dataset in Table 7 of Zuo et al. (2022b).
  - Batch size: We used same batch sizes as the optimal ones reported for each dataset in Table 7 of Zuo et al. (2022b).
  - Weight Decay: Discrete uniform over the set $\{0, 0.01, 0.1\}$
  - Distillation Regularization ($\lambda_{distill}$ in (Zuo et al., 2022b)): Discrete uniform over the set $\{1, 2, 3, 4, 5\}$.
  - $k$: 2
  - $\tau$ (for *MOESART*): 1.0.
  - $\lambda$ (for Trimmed Lasso regularization for *MOESART*): Discrete uniform over the set $\{0.0001, 0.001, 0.01, 0.1, 1.0, 10.0\}$.
  - Epochs: 10. Best model was recovered at best checkpoint based on development set.

