# OpenReview forum: "MOESART: An Effective Sampling-based Router for Sparse Mixture of Experts"
_ICLR.cc/2024/Conference — Submitted to ICLR 2024_

### Official Review · Reviewer_d6Z2 · 2023-10-26

**Soundness:** 2 fair
**Presentation:** 2 fair
**Contribution:** 2 fair
**Rating:** 5
**Confidence:** 4

**Summary:**

This work proposes MOESART, a k-sparse routing strategy for SMOE training. Instead of directly taking experts from the router's output, MOESART relies on sampling to construct a sparse and unbiased estimator of the router's softmax distribution. In the experiments, MOESART consistently outperforms several routing strategies on a wide range of tasks, ranging from recommender systems, vision, to NLP.

**Strengths:**

- This work addressing the challenge of designing a good k-approximation of the softmax, an important research problem that can impact many research topics. The proposed method is generally sound, and achieved encouraging results on a wide range of tasks.

**Weaknesses:**

## Major concern - Lack comparison with differentiable Top-k strategies

- Much of the paper compare MOESART against the standard (nondifferentiable) Top-k strategies. To give a better picture of this work's contribution, it is important to also compare MOESART against differentiable top-k strategies in both technical contributions, performances, and time complexity. The authors already mentioned the relevant baselines in Hazimeh et al., 2021; Ibrahim et al., 2023;  and Sander et al., 2023 but none are further discussed.

Furthermore, the theoretical guarantee of MOESART is also rather weak. There are other alternatives such as REINFORCE, that is unbiased but are not widely used in this setting because of its high variance.

## Major concern - Contribution of the Trimmed Lasso regularization

- How much does the Trimmed Lasso regularization contribute to the performance gains of MOESART? Does this regularization also benefit existing baselines?

 ## Minor concern - Experiment details

- In Table 1, the naive Top-k baseline seems to be the second-best method on most metrics. This result quite contradicts the literature where other advanced routing algorithms should outperform Top-K. Further investigations are needed.
- The NLP experiment in Table 2 should include X-MoE, which was originally proposed for this application.

## Minor concern - Presentation

- The proposed method is presented quite poorly in Section 3. The Trimmed Lasso regularization is a general strategy but is presented in the middle of the method discussion.
- In Section 3, it would be better to first detail the complete MOESART algorithm, then discuss its property (MOESART is a good k-sparse approximation) and the regularization used during implementation.

## Other suggestions
- It would be useful to visualize the logits and softmax distribution before and after adjustment.

**Questions:**

- What are the contributions of MOESART compared to existing differentiable Top-k in terms of technical contributions, performances, and time complexity.
- How much did the Trimmed Lasso regularization contribute to the performance gains?
- Please provide an explanation for the inconsistency results in Table 1.
- If possible, please include X-MoE in Table 2.

---

> ### Author Response · Authors · 2023-11-16
> **Rebuttal by Authors**
>
> Reviewer D6Z2
>
> We thank the reviewer for their valuable feedback and time spent reviewing the paper.
> - **Comparison with differentiable routers.**
>
>   1. **Technical Contributions**. Differentiable routers e.g., DSelect-k (Hazimeh et al., 2021) and COMET (Ibrahim et al., 2023) consider optimization-based smooth formulations based on binary representations to make routing decisions. In contrast, we consider sampling-based routing. The differentiable routing strategies e.g., DSelect-k and COMET can not do sparse training and thus can not be used to scale up model training which is a major focus of this work. These routers require computing gradients with respect to a larger set of experts per-input example while training, making them significantly more expensive. In contrast, MOESART requires gradients with respect to exactly $k$ e.g., 2 experts per-input. Although, in principle, these differentiable routers may partially allow sparse training because of their dense-to-sparse nature, the open-source implementations of these routing approaches do not support such partially conditional training and perform fully dense training. In contrast, our proposed routing approach MOESART completely allows sparse training and can easily substitute existing sparse implementations of Top-$k$ style approaches.
>
>   2. **Performance comparison** We have run new experiments to compare our routing approach with some state-of-the-art differentiable routers. These results are added in the revised draft of the paper in Supplement (Section S4). We compare across various datasets and report the performance metrics below. MOESART appears to be quite competitive with differentiable routers. Notably, MOESART even outperforms differentiable routers across many tasks.
>
>       a. *Recommender systems*:
>
> | Dataset | Router | Test Loss ($\times 10^{-2}$)  $\downarrow$  | Task-1 AUC  $\uparrow$      | Task-2 MSE $\downarrow$ |
> |------------------------------|-----------------|---------------------------------|-------------------------------|-----------------------------------|
> | Books (TW=(0.1,0.9))       | DSelect-$k$   | $241.64\pm0.14$                 | $59.05\pm0.14$                | $2.615\pm0.002$                   |
> |                                          | COMET          | $\textbf{241.13}\pm0.11$    | $\textbf{66.13}\pm0.09$   | $\textbf{2.612}\pm0.001$      |
> |                                          | MOESART     | $242.47\pm0.09$                 | $64.99\pm0.13$                | $2.626\pm0.001$                  |
> | Books (TW=(0.9,0.1))        |  DSelect-$k$  |  $~~74.22\pm0.03$              | $77.61\pm0.02$                | $\textbf{2.632}\pm0.002$    |
> |                                           | COMET          | $~~74.29\pm0.03$              | $77.30\pm0.02$                | $2.636\pm0.002$                  |
> |                                           | MOESART     | $~~\textbf{73.68}\pm0.02$ | $\textbf{78.03}\pm0.03$   | $2.641\pm0.003$                  |
> | MovieLens (TW=(0.1,0.9)) |  DSelect-$k$  | $~~73.81\pm0.03$              | $\textbf{87.79}\pm0.03$   | $0.7715\pm0.0003$              |
> |                                           | COMET          | $~~74.19\pm0.03$              | $87.67\pm0.07$                | $0.7756\pm0.0004$               |
> |                                           | MOESART     | $~~\textbf{73.60}\pm0.02$ | $87.33\pm0.03$                | $\textbf{0.7684}\pm0.0002$  |
> | MovieLens (TW=(0.9,0.1)) | DSelect-$k$   | $~~41.40\pm0.03$              | $91.44\pm0.01$                | $0.7582\pm0.0005$                |
> |                                           | COMET          | $~~41.25\pm0.04$              | $91.45\pm0.02$                | $0.7563\pm0.0008$                |
> |                                           | MOESART     | $~~\textbf{40.89}\pm0.02$ | $\textbf{91.61}\pm0.01$   | $\textbf{0.7430}\pm0.0003$   |
>
>       b. Image Datasets:
> | Dataset | Router | Test Loss ($\times 10^{-2}$) $\downarrow$ | Task-1 Accuracy $\uparrow$ | Task-2 Accuracy $\uparrow$ |
> |---------------|----------|--------------|--------------|---------------|
> | Multi-MNIST | DSelect-$k$ | $~~6.93\pm0.06$ | $98.14\pm0.02$ | $97.68\pm0.03$ |
> |          | COMET | $~~6.83\pm0.06$ | $98.21\pm0.02$ | $97.63\pm0.03$ |
> |          | MOESART | $~~\textbf{5.86}\pm0.03$ | $\textbf{98.40}\pm0.02$  | $\textbf{97.92}\pm0.02$ |
> | Multi-FashionMNIST | DSelect-$k$ | $~36.88\pm0.21$ | $87.37\pm0.07$  | $86.61\pm0.09$ |
> |          | COMET | $~34.88\pm0.13$ | $87.97\pm0.06$   | $87.42\pm0.06$ |
> |          | MOESART | $~\textbf{32.85}\pm0.11$   | $\textbf{88.56}\pm0.06$ | $\textbf{88.02}\pm0.07$ |
>
> - References
>   - Hussein Hazimeh, Zhe Zhao, Aakanksha Chowdhery, et al. DSelect-k: Differentiable selection in the mixture of experts with applications to multi-task learning. NeurIPS, 2021.
>   - Shibal Ibrahim, Wenyu Chen, Hussein Hazimeh, et al. Comet: Learning cardinality constrained mixture of experts with trees and local search. KDD, 2023.

---

> > ### Author Response · Authors · 2023-11-16
> > **Rebuttal by Authors**
> >
> > - **Contribution of the Trimmed Lasso regularization** Ablation study showing the performance of MOESART without and with trimmed lasso regularization is given in Table S2 in Supplement Section S3.3. In some cases, e.g., Books dataset, we observe substantial performance improvement with trimmed lasso regularization. In other cases e.g., MovieLens, we observe smaller improvement with trimmed lasso regularization. Recall that $g(x)$ is not sparse by construction in MOESART. Sampling injects sparsity in $\tilde{g}(x)$ during training. The goal of trimmed Lasso regularization is to encourage $g(x)$ to accumulate router probability mass per-input $x$ in the top $k$ elements, making the sampling process more deterministic per-input for inference by the end of training. Given that other Top-$k$ style routers are sparse by construction, it does not make sense to augment those routers with trimmed lasso regularization.
> > - **Trend in results in Table 1**. The reviewer makes an interesting observation that Top-$k$ seems to mostly outperform advanced Top-$k$ strategies (e.g. V-MoE, SMoE, X-MoE, Expert Choice Routing) in our experiments. We shed some light on why these advanced routing strategies do not seem to outperform Top-$k$ across tasks.
> >
> >   - Although some initial Sparse MoE literature (Shazeer et al., 2017; Fedus et al., 2022) has proposed the use of noise in Top-k, there has been some follow up work by Zoph et al., 2022 that highlights degradation in performance with noise/jitter --- see Table 3 in Zoph et al. (2022).
> >
> >   - The authors of Expert Choice Routing (Zhou et al. (2022)) replace every other Feedforward layer by an MoE layer in their transformer architectures for their experiments. Given that Expert Choice Routing applies softmax operation along the sample dimension, it can result in data leakage, which can be problematic for decoder components for encoder-decoder architectures considered by Zhou et al. (2022). Additionally, it maybe possible that some of the gains reported by Expert Choice Routing (Zhou et al., 2022) over Switch Transformers perhaps come from differences in k i.e., Switch Transformer (with $k=1$) was compared against Expert Choice Routing (with $k=2$), under same overall compute budget.
> >
> > - **X-MoE Results for Table 2** We would like to highlight that authors of X-MoE (Chi et al., 2022) didn’t consider distillation of a pre-trained non-MoE model to an MoE model in their experiments. However, on reviewer’s request, we have run new experiments with X-MoE router in this setup and we report the numbers below on GLUE benchmarks. We observe some performance gain of X-MoE over Top-$k$ router on two tasks, however X-MoE doesn’t appear to consistently outperform Top-$k$, which appears to be consistent with our experiments on recommender systems and image datasets in Table 1. MOESART consistently outperforms X-MoE on these benchmarks as well.
> >
> > |             | RTE         | CoLA         | MRPC       | SST-2       | QNLI     | QQP      | MNLI       |
> > |---------|------------|--------------|----------------|-------------|----------|-----------|------------|
> > |             | Acc $\uparrow$   | Mcc $\uparrow$ | F1 $\uparrow$    | Acc $\uparrow$ | Acc $\uparrow$   | F1 $\uparrow$ | m/mm $\uparrow$  |
> > | MoEBERT with Top-$k$     | 68.95    | 57.86  | 86.76    | 92.78    | 91.63  | 88.10   | 85.14  |
> > | MoEBERT with X-MoE       | 67.51     | 55.48   | 86.76   | 92.55    | 91.73   | 88.25    | 85.08  |
> > | MoEBERT with MOESART | $\textbf{70.40}$     | $\textbf{58.20}$ | $\textbf{88.24}$     | $\textbf{93.12}$ | $\textbf{91.95}$ | $\textbf{88.27}$ | $\textbf{85.21}$ |
> >
> > - **Presentation** We appreciate the reviewer for the suggestion to reorganize Section 3. Following the reviewer’s suggestion, we have deferred the discussion on Trimmed Lasso to a separate section (Section 3.2 in the revised draft) after the complete description of MOESART (sampling-based loss approximation and reweighting).
> >
> > Hope our new results, clarifications and revision addressed the reviewer’s comments and concerns.
> >
> > - References
> >   - Zewen Chi, Li Dong, Shaohan Huang, et al. On the representation collapse of sparse mixture of experts. NeurIPS, 2022.
> >   - William Fedus, Barret Zoph, and Noam Shazeer. Switch transformers: Scaling to trillion parameter models with simple and efficient sparsity. JMLR, 2022.
> >   - Noam Shazeer, *Azalia Mirhoseini, *Krzysztof Maziarz, et al. Outrageously large neural networks: The sparsely-gated mixture-of-experts layer. ICLR, 2017.
> >   - Yanqi Zhou, Tao Lei, Hanxiao Liu, et al. Mixture-of-experts with expert choice routing. NeurIPS, 2022.
> >   - Barret Zoph, Irwan Bello, Sameer Kumar, et al. St-moe: Designing stable and transferable sparse expert models. 2022.

---

> > > ### Comment · Reviewer_d6Z2 · 2023-11-17
> > > **Remaining Concerns**
> > >
> > > I appreciate the Authors' efforts in addressing my concerns.
> > > - However, my concerns regarding the Trimmed Lasso regularization remains. In Table S2, it seems like in some cases, the performance gain from MOESART largely comes from this regularization strategy.  Moreover, Table S2 also reports the results of the best run over 500 cross-validation trials. How sensitive the performance is with respective to $\lambda$? If extensive efforts are required to tune $\lambda$, this might be unattractive for large scale applications in LLM, which is the original and most promising applications of SMoE.
> > >
> > > - The reason for not applying the trimmed lasso regularization to other strategies is also not convincing. Given that $g(x)$ is the same for all methods, trimmed lasso is applied on $g(x)$, and then different strategies sparsify $g(x)$, it is unclear why trimmed lasso is not compatible with SMoE.
> > >
> > > - The authors haven't clarified the significance of MOESART's theoretical property.

---

> ### Author Response · Authors · 2023-11-20
> **Rebuttal by Authors for remaining concerns**
>
> We thank review for engaging with us in the discussion period and their insightful comments.
>
> - **Sensitivity to Hyperparameters.** We provide a new ablation study in Supplement Section S5 to highlight how MOESART can achieve the same level of performance as Top-k with a much lesser number of trials. This is shown in Figure S2 (in revised draft). We observe tuning reduction by a factor of $\sim100\times$ for MOESART over Top-k routing. Alternatively, a few trials 2-10 can give a substantial gain with MOESART over Top-k router, which indicates that MOESART is not too heavily dependent on a very restricted set of hyperparameter values to give improvement in performance.
> - **Use of trimmed lasso for other strategies e.g., Top-$k$** There may have been a misunderstanding, which we would like to clarify.
>     When we noted that "Top-$k$ styler routers are sparse by construction" and the use of trimmed lasso does not seem appropriate, we meant the following formulation:
>     $$\min \hat{E} [\ell (y, \sum_i f_i (x) g(x)_i )) + \lambda TrimmedLasso(g(x))] ~~~\text{(a)}$$
> where $g(x)$ is parameterized as $Softmax(Topk(Ax+b,k))$, and trimmed lasso penalty penalizes the smallest $n-k$ elements of $g$ per-input $x$. Given that the smallest $n-k$ entries of the above $g(x)$ are 0, the trimmed lasso penalty has no effect.
> Perhaps, the reviewer is suggesting the following formulation:
>     $$\min \hat{E} [\ell (y, \sum_i f_i (x) g(x)_i )) + \lambda TrimmedLasso(Softmax(Ax+b))] ~~~\text{(b)}$$
> where $g(x)$ is parameterized as $Softmax(Topk(Ax+b,k))$. We ran new experiments to test this second formulation (b) on Books dataset, where we tuned over the trimmed lasso penalty. We did not observe any performance gain by imposing the trimmed lasso regularization when considering Top-$k$ router. We show the results below:
>
> | Dataset | Router | Reg. | Test Loss ($\times 10^{-2}$) $\downarrow$ |
> |-----|------|-------|-------|
> | Books  | Top-$k$ | w/o Trimmed Lasso | $\textbf{247.00}$ |
> |       |  Top-$k$ | w/ Trimmed Lasso | $250.48$ |
> |       | MOESART | w/o Trimmed Lasso | $246.50$ |
> |       | MOESART | w/ Trimmed Lasso | $\textbf{243.02}$ |
>
> Interestingly, we observe a decrease in performance with trimmed lasso for Top-$k$. This maybe because trimmed lasso can further accelerate the probability mass into the top-$k$ elements that are being already selected by the router --- this may reduce the exploration capability of Top-$k$, affecting performance. These experiments further confirm the usefulness of imposing Trimmed Lasso regularization for our sampling-based approach.
>
> - **Theoretic rigor of MOESART**
>   - Our work has a similar rigor as other Sparse MoE papers that have practically shown benefits of Sparse MoE e.g., (Shazeer et al., 2017; Fedus et al., 2022; Zoph et al., 2022; Zhou et al., 2022; Chi et al., 2022). The theoretical properties of Top-k routing were not shown for any of these methods in literature. Interestingly, theoretical properties of vanilla Top-k routing in Gaussian MoE have started to be investigated recently, e.g., see a concurrent ICLR submission (https://openreview.net/forum?id=jvtmdK69KQ). Hence, studying theoretic properties of sparse routing methods is an open research area and require independent investigations.
>   - Our motivation to consider the MOESART objective stems from the following goal: We aim to sparsely approximate the dense-softmax-based MoE. In order to show evidence for this goal, we motivate in terms of smaller bias:
>     - The loss distribution across samples of MOESART tends to be close to dense softmax-based classical MoE objective — this is shown in Figure 2. Given that the loss distribution is similar, this can suggest that the gradients have relatively low bias.
>     - The adjusted router weights  $\tilde{g}$ (based on eq (3) in paper) after sampling have a smaller bias i.e., they are close in expectation to the dense router weights $g$,
> $$ ||(E[\tilde{g}_1],\cdots,E[\tilde{g}_n])-(g_1,\cdots,g_n)||_2~~~\text{is small}$$
> This is shown to be true with an ablation study in Figure S1 in Supplement Section S3.1.
>
> These explanations show good justification for why MOESART is effective for routing.
>
> - **REINFORCE-style estimators** As reviewer notes, it has been pointed out in literature that REINFORCE-style estimators tend to have prohibitively high variance. It is especially true for networks that have other sources of randomness (i.e., dropout or other independent random variables). Despite attempts aimed at reducing variance of such estimators, such estimators have failed to work in MoE training — see Kool et al. (2021).
>
> After the new experiments and clarifications, we would like to ask the reviewer to kindly revisit their evaluation and consider increasing the score.
> - References
>   - Wouter Kool, Chris J. Maddison, and Andriy Mnih. Unbiased gradient estimation with balanced assignments for mixtures of experts. In I (Still) Can’t Believe It’s Not Better! NeurIPS 2021 Workshop, 2021.

---

> ### Comment · Reviewer_d6Z2 · 2023-12-04
> **Acknowledgement**
>
> I appreciate the Authors' efforts in addressing my concerns. After reviewing the revision and the discussions, although my view towards this work is more positive, I cannot give a strong support for accepting this work.
>
> In the following, I highlight the key strengths and remaining weaknesses of this work.
>
> ### Strengths
> - The method is technically correct and provide consistent improvements over existing baselines
>
> ### Weaknesses
> - The improvements, although consistent, are quite marginal (mostly $1-2\\%$). Since the paper mostly focuses on small scale experiments, it is unclear how promising the proposed method would be on more challenging, large scale datasets. I understand that the limited computing resources is a challenge, but I think that the authors could explore other challenging applications of SMOE (such as continual learning [A]) to demonstrate the practical contribution of this work.
> - The theoretical results are not very strong.
>
> [A] Caccia, Lucas, et al. "On anytime learning at macroscale." Conference on Lifelong Learning Agents. PMLR, 2022.

---

### Official Review · Reviewer_7cYf · 2023-10-30

**Soundness:** 3 good
**Presentation:** 3 good
**Contribution:** 2 fair
**Rating:** 5
**Confidence:** 3

**Summary:**

MOESART introduces an MoE router sampling and expert weighting strategy which they show empirically, learns a better sparse approximation of the classical (dense) MoE models than Top-k style routers while still being sparse.

**Strengths:**

The proposed method outperforms other methods while still being sparse.

**Weaknesses:**

- The paper uses top-k != 1 for all their experiments and doesn't show results for k=1.
- The paper shows results on relatively small datasets / training setups.

**Questions:**

Is the paper saying it introduces "weighting expert outputs by output of router (aka sampling weight)"? I've seen this done in multiple MoE implementations.

---

> ### Author Response · Authors · 2023-11-16
> **Rebuttal by Authors**
>
> We thank the reviewer for their comments and time spent reviewing the paper.
>
> - **k!=1** Our motivation to consider k > 1 was based on earlier work (Du et al., 2022; Zhou et al., 2022; Zoph et al., 2022; Shazeer et al., 2017; Lepikhin et al., 2021; Ruiz et al., 2021; Hazimeh et al., 2021). They consider k > 1 for good trade-off between predictive performance and the training/serving efficiency of the model. Note that for k = 1, as is the case for Top-k (Shazeer et al., 2017; Ruiz et al., 2021), the gradient with respect to the router parameters is zero. Hence, we do not run any experiments for k = 1.
>
> - **Medium/Relatively small training setups** The training setups we considered have been considered previously in some sparse mixture of experts literature (Zuo et al., 2022; Hazimeh et al., 2021; Ibrahim et al., 2023). Our experiments were run on academic-level
> compute resources. We are limited to a GPU cluster where our resources are capped at 4 V100 Tesla GPUs with 12-hour time-limits per job. Hence, we do not have the compute bandwidth to run pre-training on 100 billion parameter models, which are typically trained on 100-1000s of GPUs. Within the compute budget, we have still considered diverse models/datasets such as medium sized models e.g., MoEBERT on GLUE and SQuAD benchmarks.
>
> - **Clarification** It appears that there might be a confusion about “weighting expert outputs by output of router”. It is true that all routers have some router weights that combine expert outputs, which is a standard MoE learning paradigm. However, we propose a novel adjustment of the weights after we sample from $g$. This adjustment is discussed in eq (3). This adjustment reduces the bias of the adjusted weights $\tilde{g}$ in expectation $||{(E[\tilde{g}_1],\cdots,E[\tilde{g}_n])-(g_1,\cdots,g_n)}||_2$. The reduction in bias is shown in ablation study in Figure S1 in Supplement Section S3.1.
>
> Hope our clarifications addressed the reviewer’s comments and concerns.
>
> - References
>   - Nan Du, Yanping Huang, Andrew M Dai, et al. GLaM: Efficient scaling of language models with mixture-of-experts. ICML 2022.
>
>   - Hussein Hazimeh, Zhe Zhao, Aakanksha Chowdhery, et al. DSelect-k: Differentiable selection in the mixture of experts with applications to multi-task learning. NeurIPS, 2021.
>
>   - Shibal Ibrahim, Wenyu Chen, Hussein Hazimeh, et al. Comet: Learning cardinality constrained mixture of experts with trees and local search. KDD, 2023.
>
>   - Dmitry Lepikhin, HyoukJoong Lee, Yuanzhong Xu, et al. GShard: Scaling giant models with conditional computation and automatic sharding. ICLR, 2021.
>
>   - Carlos Riquelme Ruiz, Joan Puigcerver, Basil Mustafa, et al. Scaling vision with sparse mixture of experts. NeurIPS, 2021.
>
>   - Noam Shazeer, *Azalia Mirhoseini, *Krzysztof Maziarz, et al. Outrageously large neural networks: The sparsely-gated mixture-of-experts layer. ICLR, 2017.
>
>   - Yanqi Zhou, Tao Lei, Hanxiao Liu, et al. Mixture-of-experts with expert choice routing. NeurIPS, 2022.
>
>   - Barret Zoph, Irwan Bello, Sameer Kumar, et al. St-moe: Designing stable and transferable sparse expert models. 2022.
>
>   - Simiao Zuo, Qingru Zhang, Chen Liang, et al. MoEBERT: from BERT to mixture-of-experts via importance-guided adaptation. NAACL: Human Language Technologies, 2022. ACL.

---

### Official Review · Reviewer_Fbe3 · 2023-11-01

**Soundness:** 2 fair
**Presentation:** 2 fair
**Contribution:** 2 fair
**Rating:** 5
**Confidence:** 4

**Summary:**

The author studied the sampling of MoE models. Specifically, the author proposes a sampling-based routing mechanism, which samples the expert index from a distribution originating from the softmax distribution. The author conducts empirical study on various datasets, and the results support the effectiveness of the proposed method.

**Strengths:**

1. The studied problem (expert router training for MoE models) is important and may have a big impact.
2. The proposed method demonstrates consistent performance gain on various benchmarks over (Top-k, V-MoE, Expert Choice Router, and X-MoE).

**Weaknesses:**

My major concern with this study is that the rationale of the proposed method is not clear. The proposed method is complicated and is only assessed with empirical comparisons. Ablation studies play a crucial role in assessing the effectiveness of the proposed method but are only included in the ablation study briefly. Particularly, some questions that I have include:
1. why it is desired to use a sampling-based expert index assignment? Figure 2 suggests that MOESART has a smaller gap between training and inference. However, it is unclear to me whether such a smaller gap is a result of the introduced regularization.
2. why it is desired to update the scaling factor as proposed? As mentioned by the author, the gradient of the expert sampling is typically ignored by MoE training (but can be estimated with the policy gradient method), and the router training is conducted with only the scaling factor. Intuitively, modifying the scaling factor in a way could lead to better gradient estimation and thus better MoE training. However, it is unclear how this is achieved by the proposed method.

Another major concern is the choice of baseline. Specifically, as in the Switch Transformer paper (in the appendix), the author suggests that softmax-based expert index sampling does not perform well. Instead, the Switch Transformer paper proposes to use jitter noise. Compared to the V-MoE method, the jitter noise proposed in Switch Transformer samples multiplicative random noise from a uniform distribution, which leads to sparse expert sampling during training. As the focus of this study is also sparse and the impact of the Switch Transformer study, I believe it is necessary to include the Switch Transformer / jitter noise as a baseline.

**Questions:**

In the current draft, the proposed method is positioned as a sampling method. But from my understanding, the sampling process of the expert index remains unchanged from the softmax sampling. Instead, the proposed method focuses on introducing additional regularization and changing expert output scaling. I would suggest the author change the wording like "carefully designed sampling".

---

> ### Author Response · Authors · 2023-11-16
> **Rebuttal by Authors**
>
> We thank the reviewer for their comments and time spent reviewing the paper.
>
> - **Clarification** Sampling based expert assignment can have a smaller bias as it gives more room for exploration amongst experts compared to Top-k routing. In Figure 2, the trimmed lasso regularization was set to 0 to particularly highlight the usefulness of the sampling based routing scheme without any regularization. We have clarified this in the figure caption in the revised draft.
>
> - **Scaling**  We adjust/scale the weights after we sample from $g$. This adjustment, outlined in eq (3), can reduce the bias of the adjusted weights $\tilde{g}$, which can be measured in expectation as $||{(E[\tilde{g}_1],\cdots,E[\tilde{g}_n])-(g_1,\cdots,g_n)}||_2$.
> We study this bias in ablation study in Section S3.1 in the Supplement. In this ablation study we consider the bias for eq (3) for different choices for $g$: uniform, random, etc. We show the plots in Fig. S1. For uniform setting, we observe the bias to be very similar to some alternative choices. However, for sufficiently non-uniform distributions, we can observe that there can be a significant reduction in bias with the proposed adjustment strategy in eq (3). In Sparse MoE, the distribution $g(x)$ can be very different across different inputs $x$, hence our proposed strategy is expected to have a lower bias overall across samples. A smaller bias can lead to improved performance. We observe this to be the case in Sparse MoE, which we show in Table S1 in the ablation study in Supplement Section S3.2.
>
> - **Gradient Estimation with policy gradient method** It has been shown in Kool et al. (2021) that estimators based on policy gradient do not work well in Sparse MoE training.
>
> - **Comparison with SMoE router e.g., Top-k with multiplicative jitter** We have run new experiments to compare against SMoE with multiplicative noise/jitter from Fedus et al., 2022. We do not see any substantial improvement of SMoE over Top-k or V-MoE. In
> fact, we observe the performance can sometimes degrade. Our MOESART significantly outperforms SMoE router as well. We have updated Table 1 in the revised draft to include SMoE numbers.
>
> - References
>   - William Fedus, Barret Zoph, and Noam Shazeer. Switch transformers: Scaling to trillion parameter models with simple and efficient sparsity. JMLR, 2022.
>
>   - Wouter Kool, Chris J. Maddison, and Andriy Mnih. Unbiased gradient estimation with balanced assignments for mixtures of experts. In I (Still) Can’t Believe It’s Not Better! NeurIPS 2021 Workshop, 2021. URL https://openreview.net/forum?id=Hvfva7l1tcj.

---

> ### Author Response · Authors · 2023-11-16
> **Rebuttal by Authors**
>
> - **Sampling based variant of SMoE** We appreciate the reviewer for bringing this to our attention that Fedus et al. (2022) considered a sampling version of SMoE in their Appendix for one experiment. In this experiment, they showed that the sampling variant of SMoE can degrade performance compared to top-k based SMoE. We have run new experiments to compare against this variant of SMoE (denoted by SMoE-S) and we provide the results below. We observe a mixed trend, where SMoE-S sometimes outperforms SMoE. In comparison, MOESART consistently outperforms SMoE and SMoE-S.
>
>   a) Recommender Systems:
> | Dataset  | Router   | Test Loss ($\times 10^{-2}$) $\downarrow$ | Task-1 AUC $\uparrow$  | Task-2 MSE $\downarrow$   |
> |-----------|----------|---------|---------|-------|
> | Books (TW=(0.1,0.9))    | SMoE   | $249.88\pm0.13$ | $56.68\pm0.09$  | $2.707\pm0.001$ |
> |               | SMoE-S       | $251.69\pm0.17$ | $53.86\pm0.07$   | $2.727\pm0.002$ |
> |               | MOESART    | $\textbf{242.47}\pm0.09$   | $\textbf{64.99}\pm0.13$ | $\textbf{2.626}\pm0.001$ |
> | Books (TW=(0.9,0.1)) | SMoE  | $~~75.30\pm0.05$  | $77.13\pm0.05$  | $2.718\pm0.002$ |
> |               | SMoE-S        | $~~75.88\pm0.06$ | $77.63\pm0.06$ | $2.824\pm0.003$ |
> |               | MOESART    | $~~\textbf{73.68}\pm0.02$ | $\textbf{78.03}\pm0.03$ | $\textbf{2.641}\pm0.003$ |
> |MovieLens (TW=(0.1,0.9)) | SMoE | $~~79.47\pm0.07$ |  $85.36\pm0.06$ |  $0.8303\pm0.0008$ |
> |               | SMoE-S       | $~~73.89\pm0.02$ | $85.89\pm0.02$  | $0.7691\pm0.0002$ |
> |               | MOESART  | $~~\textbf{73.60}\pm0.02$ | $\textbf{87.33}\pm0.03$  | $\textbf{0.7684}\pm0.0002$ |
> | MovieLens (TW=(0.9,0.1))   |  SMoE | $~~43.08\pm0.06$ | $90.86\pm0.03$ | $0.7917\pm0.0009$ |
> |               | SMoE-S      | $~~41.17\pm0.02$ | $91.58\pm0.01$  | $0.7543\pm0.0004$ |
> |                | MOESART  | $~~\textbf{40.89}\pm0.02$ | $\textbf{91.61}\pm0.01$  | $\textbf{0.7430}\pm0.0003$ |
>
>   b) Image Datasets
> | Dataset | Router  | Test Loss ($\times 10^{-2}$) $\downarrow$ | Task-1 Accuracy $\uparrow$ | Task-2 Accuracy $\uparrow$ |
> |----------|----------|---------|----------|------------|
> | Multi-MNIST | SMoE | $~~6.90\pm0.05$ | $98.18\pm0.02$ | $97.69\pm0.02$ |
> |             |SMoE-S     | $~~7.19\pm0.09$ | $98.11\pm0.04$ | $97.45\pm0.05$ |
> |             |MOESART | $~~\textbf{5.86}\pm0.03$ | $\textbf{98.40}\pm0.02$ | $\textbf{97.92}\pm0.02$  |
> |Multi-FashionMNIST     | SMoE | $~34.68\pm0.09$ | $88.06\pm0.04$ | $87.52\pm0.06$ |
> |             | SMoE-S    | $~34.39\pm0.10$ | $88.06\pm0.08$ | $87.82\pm0.08$ |
> |             | MOESART   | $~\textbf{32.85}\pm0.11$ | $\textbf{88.56}\pm0.06$ | $\textbf{88.02}\pm0.07$ |
>
> - **Difference in paramaterization of SMoE-S and MOESART**.
>     Next, we describe the differences in our approach with that followed by SMoE-S. Although both approaches i.e., MOESART~and SMoE perform sampling, there are key differences in parameterization, weighting and regularization:
>
>   (i) SMoE-S considers the following parameterization: $\hat{g}(x) = g(x) \odot 1_s$, where $1_s$ denotes a vector such that only k sampled indices are non-zero that are in the set $s$. This $\hat{g}(x)$ does not obey the simplex constraint. In contrast,  $\tilde{g}(x)$ for MOESART is based on weight adjustment outlined in eq (3) and these adjusted weights always lie on the sparse simplex. The bias of SMoE-S is higher than that for MOESART, which can degrade performance of SMoE-S.
>
>   (ii) SMoE-S is stochastic during training as well as inference. In contrast, MOESART uses top-$k$ indices at inference.
>
>   (iii) There is no trimmed lasso regularization in SMoE-S. Trimmed lasso regularization can boost performance as we show in ablation study in Table S2 in Supplement Section 3.3.
>
> - **Change choice of words** Our use of the term “carefully designed” in the phrase “carefully designed sampling and weighting strategies” was primarily a reference to adjustment strategies. Hope this clarifies the term. We are happy to change the wording to be more precise in the revision.
>
> Hope our new results and clarifications addressed the reviewer’s comments and concerns.
>
> - References
>   - William Fedus, Barret Zoph, and Noam Shazeer. Switch transformers: Scaling to trillion parameter models with simple and efficient sparsity. JMLR, 2022.

---

### Official Review · Reviewer_diTt · 2023-11-01

**Soundness:** 3 good
**Presentation:** 3 good
**Contribution:** 3 good
**Rating:** 6
**Confidence:** 2

**Summary:**

This work introduces MOESTART, a sampling-based routing method for mixture of experts models. The authors address the performance issues faced by existing top-k gate variants due to the discontinuous nature of the routing problem. To maintain k-sparsity during both training and inference, the authors propose a method that incorporates sampling to emulate the results of top-k softmax gates. The proposed method demonstrates significant improvements across diverse domains, including recommender systems, vision, and NLP.

**Strengths:**

1. The motivation behind this work is reasonable, and the proposed solution is demonstrated to be unbiased, both theoretically and practically.
2. The writing and organization of the paper are of good quality, resulting in an easy-to-read presentation.
3. The authors have conducted comprehensive experiments to thoroughly validate the effectiveness of MOESTART, ensuring the reliability of the proposed method.

**Weaknesses:**

1. It should be noted that all experiments in this work were conducted on small-sized datasets, whereas existing Mixture of Experts (MoE) structures typically focus on large-scale training and inference scenarios. This difference in dataset size should be taken into consideration when evaluating the applicability of the proposed method.
2. One limitation of the method is that it only supports k>=2, which may restrict its practical value. It would be beneficial for future improvements to extend the method's applicability to lower values of k as well.

**Questions:**

1. The motivation behind MOESTART is to achieve results that closely resemble those of the top-k softmax gate. However, it is intriguing to observe that MOESTART consistently outperforms the softmax gate in all the conducted experiments. The authors should provide additional clarification or insights as to why MOESTART exhibits superior performance compared to the traditional softmax gate, despite its primary goal of approximation.

---

> ### Author Response · Authors · 2023-11-16
> **Rebuttal by Authors**
>
> We thank the reviewer for their comments and time spent reviewing the paper.
>
> - **Medium/Relatively small training setups** The training setups we considered have been considered previously in some sparse mixture of experts literature (Zuo et al., 2022; Hazimeh et al., 2021; Ibrahim et al., 2023). Our experiments were run on academic-level compute resources. We are limited to a GPU cluster where our resources are capped at 4 V100 Tesla GPUs with 12-hour time-limits per job. Hence, we do not have the compute bandwidth to run pre-training on 100 billion parameter models, which are typically trained
> on 100-1000s of GPUs for many days. Within the compute budget, we have still considered diverse models/datasets such as medium sized models e.g., MoEBERT on GLUE and SQuAD benchmarks. Interestingly, our work exposes that some of the advanced top-k
> routing strategies do not generalize well across datasets/training setups/domains. Hence, we believe that these reasonably sized settings prove to be good regimes to explore novel routing strategies.
>
> - **k=1** Our motivation to consider k > 1 was based on earlier work (Du et al., 2022; Zhou et al., 2022; Zoph et al., 2022; Shazeer et al., 2017; Lepikhin et al., 2021; Ruiz et al., 2021; Hazimeh et al., 2021). They consider k > 1 for good trade-off between predictive performance and the training/serving efficiency of the model. Hence, we do not particularly see this as a limitation in terms of practical value.
>
> - **Performance difference between Softmax vs Sampling-based MOESART** The reviewer makes an interesting observation that MOESART tends to consistently outperform traditional softmax. We observe some differentiably sparse routers e.g., DSelect-k,
> COMET can also outperform softmax routing — see Table S4 in Supplement Section S4 in the revised draft. Hence, sparsity can be beneficial on these datasets to improve generalization. Additionally, softmax routing with linear parameterization is less richer in terms of parameterization than tree-based COMET (Ibrahim et al., 2023) and may also be less richer in comparison to MOESART because sampling in MOESART may induce more nonlinearity due to sparsity. Dense nature of softmax activates all experts for all samples, which may affect the localized/specialization property as highlighted in Ramamurti & Ghosh (1998). Given that MOESART can outperform COMET, it is possible that sparsity in MOESART may induce more localized decision making, hence leading to improved performance.
>
> Hope our clarifications addressed the reviewer’s comments and concerns.
>
> - References
>   - Nan Du, Yanping Huang, Andrew M Dai, et al. GLaM: Efficient scaling of language models with mixture-of-experts. ICML 2022.
>
>   - William Fedus, Barret Zoph, and Noam Shazeer. Switch transformers: Scaling to trillion parameter models with simple and efficient sparsity. JMLR, 2022.
>
>   - Hussein Hazimeh, Zhe Zhao, Aakanksha Chowdhery, et al. DSelect-k: Differentiable selection in the mixture of experts with applications to multi-task learning. NeurIPS, 2021.
>
>   - Shibal Ibrahim, Wenyu Chen, Hussein Hazimeh, et al. Comet: Learning cardinality constrained mixture of experts with trees and local search. KDD, 2023.
>
>   - Dmitry Lepikhin, HyoukJoong Lee, Yuanzhong Xu, et al. GShard: Scaling giant models with conditional computation and automatic sharding. ICLR, 2021.
>
>   - Viswanath Ramamurti and Joydeep Ghosh. Use of localized gating in mixture of experts networks. Applications and Science of Computational Intelligence, 1998.
>
>   - Carlos Riquelme Ruiz, Joan Puigcerver, Basil Mustafa, et al. Scaling vision with sparse mixture of experts. NeurIPS, 2021.
>
>   - Noam Shazeer, *Azalia Mirhoseini, *Krzysztof Maziarz, et al. Outrageously large neural networks: The sparsely-gated mixture-of-experts layer. ICLR, 2017.
>
>   - Yanqi Zhou, Tao Lei, Hanxiao Liu, et al. Mixture-of-experts with expert choice routing. NeurIPS, 2022.
>
>   - Barret Zoph, Irwan Bello, Sameer Kumar, et al. St-moe: Designing stable and transferable sparse expert models. 2022.
>
>   - Simiao Zuo, Qingru Zhang, Chen Liang, et al. MoEBERT: from BERT to mixture-of-experts via importance-guided adaptation. NAACL: Human Language Technologies, 2022. ACL.

---

### Author Response · Authors · 2023-11-21
**To All Reviewers**

We thank all reviewers for their comments. We have posted a point-by-point response to each reviewer's comments. We have supported our responses with additional results and comparisons, where possible. The text in blue in the revised draft are edits based on reviewer's comments. Note that main and supplement are in the same document.

If you have any further questions, please let us know so that we can try to address them before the discussion period ends tomorrow on 22-Nov.

Lastly, we would like to ask the reviewers to kindly revisit their evaluation based on the clarifications and new experiments and consider increasing the score.

---

### Meta-Review · Area_Chair_GfZr · 2023-12-16

**Metareview:**

This paper proposes a sampling-based routing method for mixture of experts (MoE) models. To maintain k-sparsity during both training and inference, it proposes a method that incorporates sampling to emulate the results of top-k softmax gates. Reviewers expressed several concerns, including the small scale of the experiments, which makes results somewhat inconclusive, the focus on k>1 (which the authors seem to have justified correctly), lack of comparison with differentiable routers. Although the authors improved the paper (e.g. adding comparisons with differentiable routers), some concerns persist -- the marginal gains in the small scale experiments make it inconclusive if the method is competitive at a larger scale.

**Justification For Why Not Higher Score:**

The marginal gains in the small scale experiments make it inconclusive if the method is competitive at a larger scale.

**Justification For Why Not Lower Score:**

N/A

---

### Decision · Program_Chairs · 2024-01-16

Reject